# Selective clearance of the inner nuclear membrane protein emerin by vesicular transport during ER stress

Abigail Buchwalter[1,2,3]*, Roberta Schulte[4], Hsiao Tsai[4], Juliana Capitanio[4], Martin Hetzer[4]*

[1]Cardiovascular Research Institute, University of California, San Francisco, San Francisco, United States; [2]Department of Physiology, University of California, San Francisco, San Francisco, United States; [3]Chan Zuckerberg Biohub, San Francisco, United States; [4]The Salk Institute for Biological Studies, La Jolla, United States

**Abstract** The inner nuclear membrane (INM) is a subdomain of the endoplasmic reticulum (ER) that is gated by the nuclear pore complex. It is unknown whether proteins of the INM and ER are degraded through shared or distinct pathways in mammalian cells. We applied dynamic proteomics to profile protein half-lives and report that INM and ER residents turn over at similar rates, indicating that the INM's unique topology is not a barrier to turnover. Using a microscopy approach, we observed that the proteasome can degrade INM proteins in situ. However, we also uncovered evidence for selective, vesicular transport-mediated turnover of a single INM protein, emerin, that is potentiated by ER stress. Emerin is rapidly cleared from the INM by a mechanism that requires emerin's LEM domain to mediate vesicular trafficking to lysosomes. This work demonstrates that the INM can be dynamically remodeled in response to environmental inputs.
DOI: https://doi.org/10.7554/eLife.49796.001

**\*For correspondence:**
abigail.buchwalter@ucsf.edu (AB);
hetzer@salk.edu (MH)

**Competing interests:** The authors declare that no competing interests exist.

## Introduction

The biogenesis of roughly one-third of the cell's proteome takes place within the endoplasmic reticulum (ER) network. The ER is contiguous with the nuclear envelope (NE) membrane, a double bilayer membrane that defines the boundary of the nucleus. The NE is punctuated by nuclear pore complexes (NPCs) which control transport between the nuclear and cytoplasmic compartments. The outer nuclear membrane (ONM) and the bulk ER membrane network have a similar protein composition, including ribosomes that can be seen associated with the ONM. The inner nuclear membrane (INM), in contrast, is cloistered away from the bulk ER network by the NPC (*Figure 1A*). Proteomic analyses suggest that hundreds of proteins are selectively concentrated at the INM (*Schirmer et al., 2003*), and mutations to these proteins cause a broad array of rare pathologies (*Schreiber and Kennedy, 2013*).

As the INM is devoid of ribosomes and translocation machinery, INM proteins must be synthesized in the ONM/ER and transported into the INM. Proteins concentrate at the INM by mechanisms including diffusion followed by stable binding to a nuclear structure, such as chromatin or the nuclear lamina, or signal-mediated import through the NPC (*Katta et al., 2014*). Transport across the NPC is a major kinetic barrier to accumulation of proteins at the INM (*Boni et al., 2015*; *Ungricht et al., 2015*). While mechanisms of INM targeting have been extensively studied, it is less clear how INM proteins are targeted for degradation if misfolded, damaged, or mistargeted.

Protein folding is inefficient, and newly synthesized proteins often become terminally misfolded and require degradation (*Hegde and Zavodszky, 2019*). Mature proteins also become damaged or misfolded over time and require selective degradation and replacement. Within the ER membrane

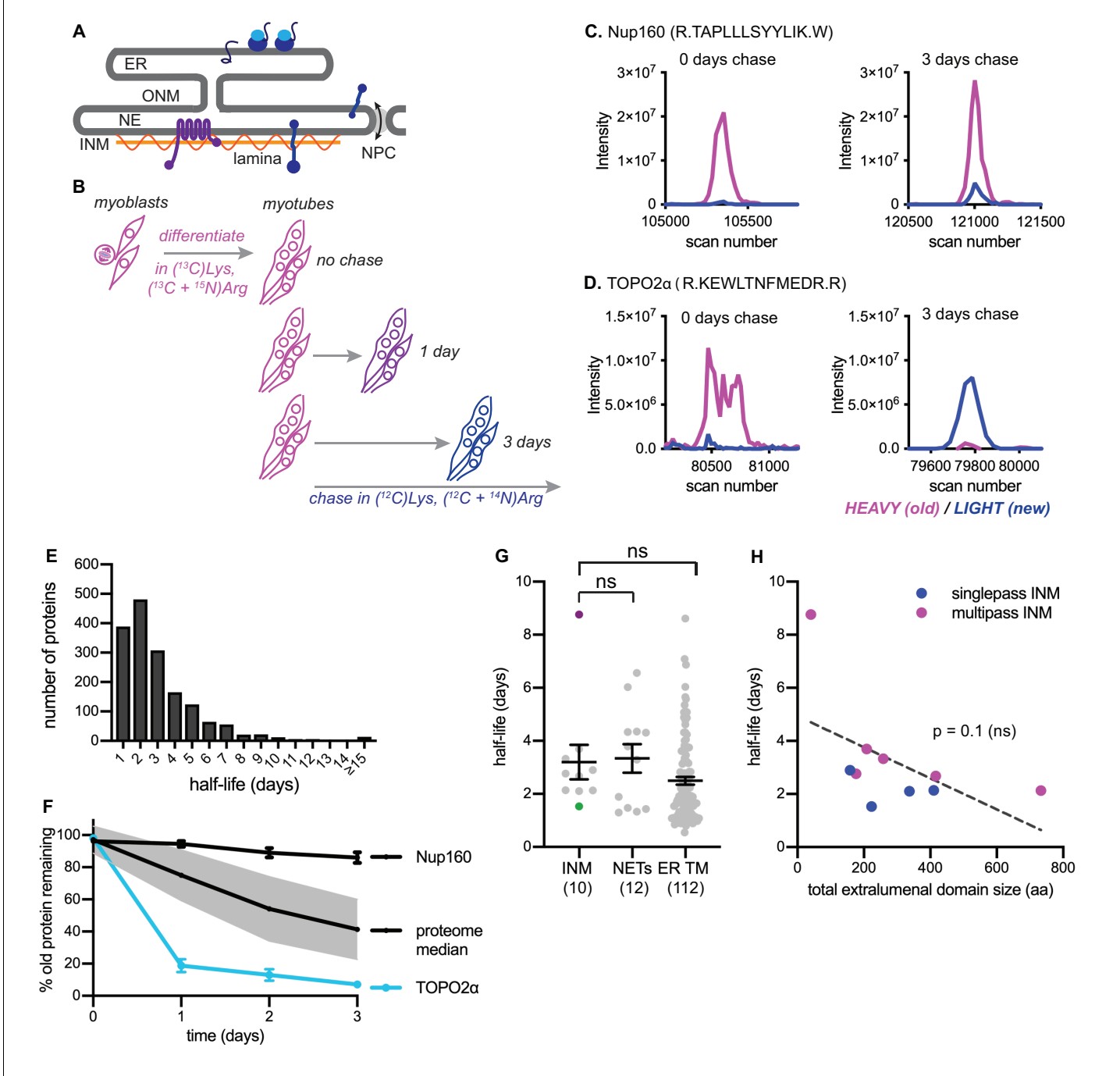

**Figure 1.** Dynamic proteomic analysis of inner nuclear membrane protein turnover. (A) Diagram of the ER with associated ribosomes, the NE composed of the ONM and INM, the NPCs, and the underlying nuclear lamina. INM proteins are synthesized in the ER, pass through the NPC, and enrich at the INM. (B) Overview of dynamic SILAC labeling experimental design. C2C12 mouse myoblasts were cultured for five population doublings in medium containing $^{13}C_6$-lysine and $^{13}C_6$, $^{15}N_4$-arginine to completely label the proteome. After 3 days of culturing under differentiating conditions to generate non-dividing myotubes, cultures were switched to chase medium containing $^{12}C$-lysine and $^{12}C$, $^{14}N$-arginine for 1 to 3 days. Nuclear extracts were prepared at day 0, day 1, day 2, and day three for proteomic identification. (C,D) Representative peptide scans for a slowly degraded protein (Nup160) and (D) for a rapidly degraded protein (Topo2α) at the starting and ending points of the experiment outlined in (B). (E) Histogram of calculated half-lives for 1677 proteins with a median half-life of 2.4 days. (F) Features of nuclear proteome turnover. Median turnover behavior of 1677 proteins detected in at least three timepoints with at least one peptide (black line) with one standard deviation (gray); compared to turnover of the slowly exchanged protein Nup160 (black) and the rapidly exchanged protein Topo2α (blue). Error bars indicate SEM. (G) Calculated half-lives of 10 *bona fide* INM proteins, ranging from slowly degraded (nurim, purple) to rapidly degraded (emerin, green); 12 nuclear envelope transmembrane proteins (NETs) identified as NE residents by subtractive proteomics (see *Schirmer et al., 2003*); and 112 ER membrane proteins. ns indicates lack of

*Figure 1 continued on next page*

*Figure 1 continued*

statistical significance by Mann-Whitney test. Error bars indicate SEM. (**H**) There is no significant correlation between extraluminal domain size of INM proteins and their half-lives. See also Source Data 1–2, *Supplementary files 1–3*, and *Figure 1—figure supplement 1*.

DOI: https://doi.org/10.7554/eLife.49796.002

The following figure supplement is available for figure 1:

**Figure supplement 1.** Example half life fits.

DOI: https://doi.org/10.7554/eLife.49796.003

network, the major degradation pathway is ER-associated degradation, or ERAD. ERAD is initiated by poly-ubiquitination of a target protein by an E3 ubiquitin ligase, followed by extraction from the membrane and proteolysis by proteasomes in the cytosol (*Hegde and Zavodszky, 2019*). Flux through ERAD helps to maintain organelle homeostasis and cell function by clearing damaged, misfolded, or mislocalized proteins.

Recent work in *S. cerevisiae* has identified a small number of ubiquitin ligases that target INM-localized proteins for degradation by ERAD, but the mammalian homologs remain elusive, perhaps because of the massive expansion of the E3 ubiquitin ligase family in recent evolution (*Deshaies and Joazeiro, 2009*). Degradation of mammalian INM proteins also appears to rely on activity of the proteasome and on the ERAD ATPase p97 (*Tsai et al., 2016*), suggesting that mammalian INM proteins may be subject to ERAD. However, we lack a broad understanding of the lifetimes of INM proteins in this compartment and the pathways used for their degradation within mammalian cells.

We sought to understand features of INM protein turnover in mammalian cells, and applied both proteome-wide and targeted candidate approaches to address this question. Here we show that the LEM domain protein emerin (EMD) is a rapidly degraded constituent of the INM. We use EMD as a model for dissecting INM protein turnover pathways and demonstrate that EMD is subject to both proteasome-dependent and lysosome-dependent modes of degradation. We report that both misfolded and normally folded variants of EMD are selectively exported from the INM and ER during acute ER stress by vesicular transport through the secretory pathway and delivery to the lysosome. These findings indicate that the INM sub-compartment senses and responds to ER stress.

## Results

### Trends in protein turnover across the NE/ER membrane network

We first used a dynamic proteomic approach to understand trends in protein turnover across ER sub-compartments. Since the nucleus is entirely disassembled during mitosis in mammalian cells, we chose a system that would allow us to profile protein turnover in the absence of cell division. We made use of the C2C12 myoblast culture system, which can be induced to irreversibly differentiate into myotubes by serum withdrawal (*D'Angelo et al., 2009*). We subjected these non-dividing mouse muscle myotubes to a pulse-chase timecourse using stable isotope labeling in cell culture (SILAC) (*Ong and Mann, 2006*) for timepoints ranging from 1 to 3 days (*Figure 1B*, see Materials and methods). Crude nuclear extracts were prepared and analyzed by mass spectrometry, and the ratio of 'old' ($^{13}C_6$-Lys, $^{15}N_4$ + $^{13}C_6$-Arg-labeled) to 'new' ($^{12}C_6$-Lys, $^{14}N_4$ + $^{12}C_6$-Arg-labeled) protein was quantified at the peptide level over time; peptides that passed stringent quality control filters were retained for estimation of half-lives by a linear regression fitting method (*Dörrbaum et al., 2018*)(see Materials and methods). We evaluated 1677 proteins and predicted half-lives over a wide range, from less than a day to greater than 15 days (*Figure 1E*, Table S3). Linear regression performs well when a line can be fitted with high fidelity and a non-zero slope is detectable; these conditions were generally met for proteins with predicted half-lives ranging from 1 to 8 days. We observed more frequent deviations in linearity at the low extreme (predicted $t^{1/2}$ <1 day) and slopes approaching zero at the high extreme (predicted $t^{1/2}$ > 8 days) (*Figure 1—figure supplement 1*). We expect that these factors limit the precision of half-life determination below 1 day and above 8 days from our 3 day timecourse. The median turnover rate that we observed (2.4 days) corresponds well with previous analyses in non-dividing mammalian cell cultures (*Cambridge et al., 2011*).

We observed some extremes in protein stability that are consistent with previous reports; for example, the long-lived nuclear pore complex component Nup160 (*Toyama et al., 2013*) was in the top 10% of predicted half-lives, with a calculated half-life of approximately 18 days (*Figure 1C,F*, Table S1). Near the other extreme, we observed that the enzyme topoisomerase 2α (Topo2α) had a predicted half-life in the bottom 10%, of less than 1 day (*Figure 1D,F*). This is consistent with this enzyme's known regulation by ubiquitination and proteolysis (*Gao et al., 2014*).

Having established this framework, we then quantified the turnover kinetics of known inner nuclear membrane (INM) proteins. For this analysis, we focused on proteins whose preferential enrichment in this membrane compartment had been experimentally verified; we identified ten such proteins in our dataset (see Table S1) and determined their half-lives (see Materials and methods). We observed half-lives for these proteins ranging from 8.8 days (nurim) to 1.5 days (emerin) (*Figure 1G*). While very little is known about nurim's function, its intrinsic biophysical properties may contribute to its long half-life: nurim contains six transmembrane domains, is extremely insoluble (*Hofemeister and O'Hare, 2005*), and diffuses very slowly within the INM (*Rolls et al., 1999*). Emerin (EMD) is a founding member of the LEM domain family of INM proteins with essential functions in muscle development (*Brachner and Foisner, 2011*). Unlike nurim, EMD is a small (~25 kDa), single-pass, tail-anchored transmembrane protein that diffuses freely through the NPC and enriches at the INM by virtue of its affinity for lamin A (*Vaughan et al., 2001*).

Given the INM's status as a restricted sub-compartment of the ER, we reasoned that it might be possible that INM proteins would be generally less accessible to protein turnover than ER membrane proteins. Alternatively, similarly effective turnover in both compartments might support the possibility that turnover can occur in situ at the INM. ER membrane proteins were well represented in our dataset, as a significant proportion of ER membranes remain attached to and co-purify with nuclei (*Schirmer et al., 2003*). We could thus query whether INM proteins exhibited distinct turnover kinetics from membrane proteins of the bulk ER by comparing INM and ER transmembrane protein half-lives. We also compared *bona fide* INM proteins to proteins that had been identified as preferentially associated with either the inner or outer bilayer of the nuclear envelope (NE) membrane by comparative proteomics (*Schirmer et al., 2003*), termed NE transmembrane proteins (NETs). Altogether, these analyses indicate that INM proteins do not exhibit unique turnover kinetics as a protein class, compared to ER membrane proteins in general or to the overlapping designation of NETs (*Figure 1G*).

The size of INM proteins determines whether a protein must rely on signal-mediated transport through the NPC (*Katta et al., 2014*), and live imaging assays indicate that INM proteins with larger nucleoplasm-facing domains move more slowly across the NPC barrier (*Boni et al., 2015*; *Ungricht et al., 2015*). If transit across the NPC and out of the INM were a prerequisite for turnover in the bulk ER, we reasoned that turnover efficiency would also exhibit some dependence on protein size, because of the relationship between protein size and transport efficiency between the two compartments. The short half-life and small size of EMD is in line with this possibility. Our dataset of INM proteins included four single-pass INM and six multi-pass INM proteins, with total size of extra-luminal domains ranging from 40 amino acids to 733 amino acids (Table S5). If export out of the NPC were a prerequisite for INM protein turnover, we reasoned that half-life should increase as the bulk of nucleoplasm-facing domains increases. We found no evidence for such a correlation (*Figure 1H*). We infer from this analysis that other factors distinct from monomeric protein size regulate protein turnover rate. This indicates that for INM proteins, export out of the INM is not a rate-limiting step for protein turnover. Rather, this is consistent with evidence in *S. cerevisiae* (*Foresti et al., 2013*; *Khmelinskii et al., 2014*) and in mammalian cells (*Tsai et al., 2016*) that turnover of INM proteins can take place in situ at the INM.

## Recombination-induced tag exchange confirms INM protein lifetimes

We observed a wide range of half-lives for INM proteins in our proteomic analyses (*Figure 1G*), with the polytopic INM protein nurim turning over most slowly and the single-pass INM protein EMD turning over most rapidly. To directly visualize these relative differences in protein stability, we used recombination-induced tag exchange (RITE) (*Toyama et al., 2019*; *Verzijlbergen et al., 2010*) (*Figure 2A*) to perform a microscopy-based pulse-chase experiment. We expressed either nurim or EMD in a cassette encoding two C-terminal epitope tags separated by LoxP sites and by a stop codon, such that the resulting transcript will encode a protein that will be C-terminally tagged with

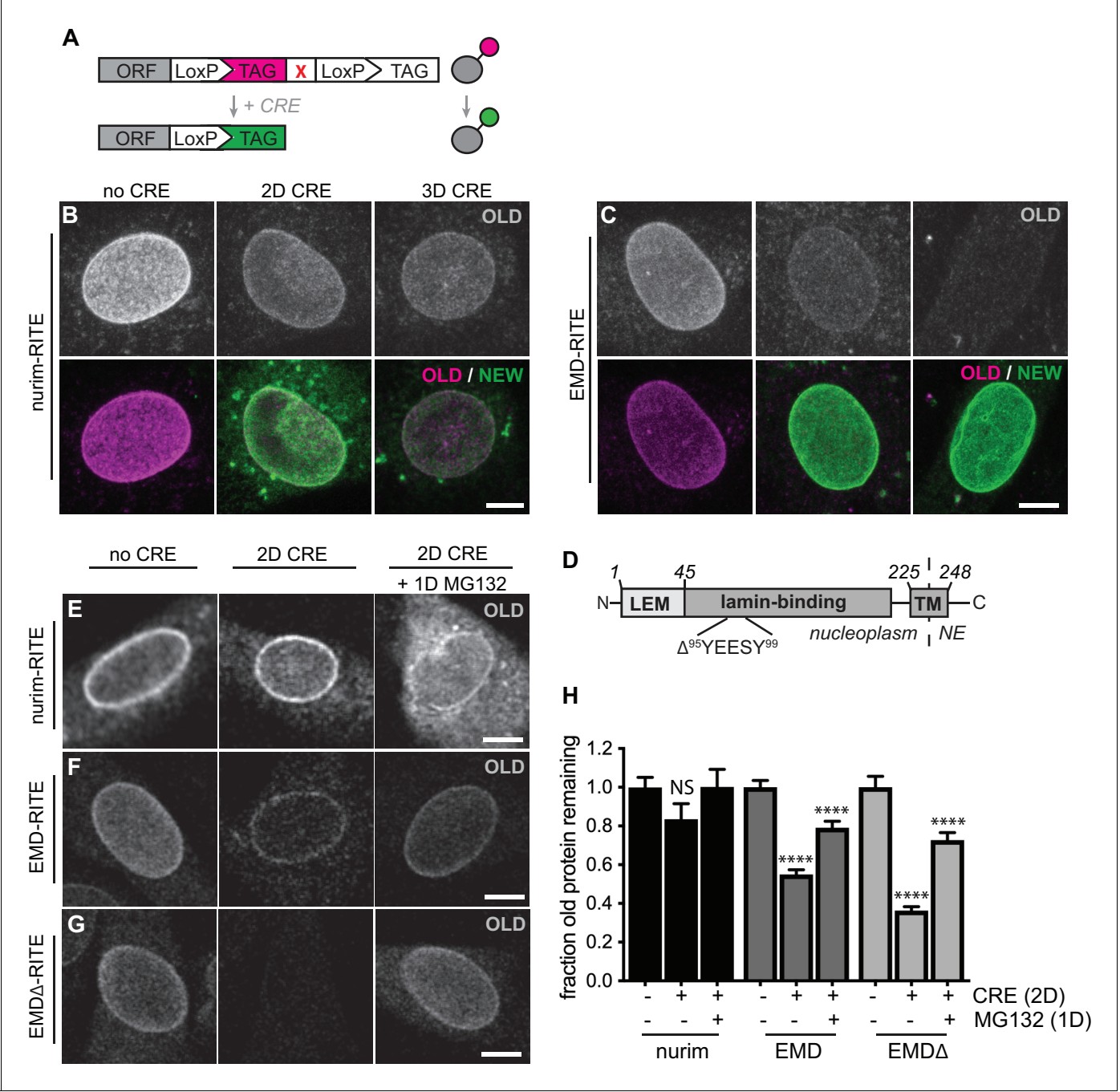

**Figure 2.** RITE analysis of INM proteins enables visualization of proteasome-dependent turnover. RITE analysis of INM proteins corroborates protein turnover determined by proteomics. (**A**) Schematic of recombination-induced tag exchange (RITE) expression cassette for visualizing protein turnover using Cre recombinase-mediated tag switching. (**B-C**) RITE timecourses of nurim-RITE (**B**) and EMD-RITE (**C**) in quiescent C2C12 cells. Maximum intensity projections of confocal z-series shown. (**D**) Diagram of emerin domain organization and position of EDMD-linked deletion mutant (EMDΔ95-99) within the lamin-binding domain. (**E-G**) RITE timecourses of nurim-RITE (**E**), EMD-RITE (**F**), and EMDΔ-RITE (**G**) with or without 1 day of cotreatment with the proteasome inhibitor MG132 (right panels). Single confocal z-slices shown. (**H**) Quantification of normalized intensity of old NE-localized RITE-tagged protein in maximum intensity projections of confocal z series acquired across the conditions shown in (**E-G**). Bars indicate average values with error bars indicating SEM for N > 42 cells per condition from 2 independent experiments. **** indicates p-value < 0.0001 (by t-test) for comparison between untreated and treated conditions. Scale bar, 10 mm. See also *Figure 2—figure supplement 1*.

DOI: https://doi.org/10.7554/eLife.49796.004

The following figure supplement is available for figure 2:

**Figure supplement 1.** Identification of a disease-linked emerin mutant with normal NE localization.

*Figure 2 continued on next page*

*Figure 2 continued*

DOI: https://doi.org/10.7554/eLife.49796.005

the first tag. Upon adenoviral introduction of Cre recombinase, the RITE cassette is recombined to remove the first tag and position the second tag downstream of the open reading frame, so that all newly synthesized mRNA encodes a protein marked with the second tag. This enables simultaneous tracking of older and newer pools of protein that were synthesized before and after Cre addition, respectively (*Toyama et al., 2019*; *Verzijlbergen et al., 2010*). Using this approach, we visualized the rate of decline in the fluorescence intensity of 'old' myc-tagged nurim or EMD over several days in quiescent C2C12 cells. Consistent with our proteomic observations, we observed that RITE-tagged nurim decayed significantly more slowly than RITE-tagged EMD at the NE (*Figure 2B–C,H*).

The RITE system allows unambiguous dissection of the fates of maturely folded protein as well as nascent, newly synthesized protein. Recent work in yeast (*Foresti et al., 2013*; *Khmelinskii et al., 2014*) and in mammalian cells (*Tsai et al., 2016*) strongly suggests that INM proteins are subject to proteasome-mediated degradation via the ERAD pathway. The RITE system provides a means to distinguish the effects of proteasome inhibition on maturely folded proteins by inhibiting the proteasome after RITE tag switching, and monitoring the effects on maturely folded proteins. Mature nurim-RITE decreases only modestly within 2 days of tag switching, but co-incubation with the proteasome inhibitor MG132 for 1 day causes accumulation of nurim-RITE through the NE and ER (*Figure 2E,H*). Maturely folded EMD-RITE diminishes significantly at the NE within 2 days of tag switching but is partially stabilized at the NE in the presence of MG132 (*Figure 2F,H*). This indicates that mature, INM-localized proteins can be degraded in a proteasome-dependent pathway in situ at the INM. Notably, abundant proteasomes have been observed along the INM in cryo-EM studies and could possibly engage with substrate there (*Albert et al., 2017*). This is also consistent with a recent report that an unstable INM protein mutant accumulates within the nucleus of mammalian cells when the proteasome is inhibited (*Tsai et al., 2016*).

## Identification of a model substrate for dissecting INM protein turnover

In order to gain more insight into the pathways that control INM protein turnover in mammalian cells, we chose to focus on EMD for its relatively fast turnover rate (*Figure 1G*, Table S1) and for the variety of disease-linked mutations to EMD that appear to influence protein stability (*Fairley et al., 1999*). Loss-of-function mutations to EMD cause Emery-Dreifuss muscular dystrophy (EDMD) (*Bonne and Quijano-Roy, 2013*). In some cases, EDMD-linked mutations cause loss of detectable EMD protein without affecting mRNA levels, suggesting that these mutations might cause misfolding and degradation of EMD (*Fairley et al., 1999*). We sought to identify such an EDMD-linked EMD variant for use as a model substrate for dissecting INM protein turnover. We selected a small in-frame deletion (Δ95–99) within EMD's lamin-binding domain (*Figure 2D*) that had been previously shown to localize to the NE when ectopically expressed (*Fairley et al., 1999*). Consistently, when we expressed either EMD-GFP or EMDΔ95–99-GFP in C2C12 cells, we observed similar enrichment at the NE (*Figure 2—figure supplement 1*). Further, both protein variants exhibited identical residence times at the NE as assayed by fluorescence recovery after photobleaching (FRAP) analysis (*Figure 2—figure supplement 1*). Directly monitoring the stability of EMDΔ95–99 by RITE tagging indicates that it disappears from the NE faster than wild type EMD (*Figure 2G*), but is also stabilized at the NE by proteasome inhibition (*Figure 2G–H*). These observations indicate that EMDΔ95–99 is an unstable EMD variant that resides within the INM. We next moved to dissect that pathway(s) involved in EMDΔ95–99 degradation.

## Proteasome-dependent and proteasome-independent modes of emerin clearance

Misfolded ER resident proteins are cleared by the ER-associated degradation (ERAD) pathway. ERAD clients are marked for degradation by ubiquitination, extracted from the ER membrane by the ATPase enzyme p97, and degraded by the proteasome in the cytosol (*Ruggiano et al., 2014*). INM proteins may also be targeted to an arm of the ERAD pathway in mammalian cells (*Tsai et al., 2016*), and our data indicate that multiple INM proteins are stabilized in situ by proteasome

inhibition. However, ubiquitin ligase(s) that recognize INM-localized substrates in mammalian cells have not been identified.

To sensitively probe factors that influence INM protein stability, we tracked the stability of GFP-tagged EMD variants. When de novo protein synthesis was blocked by cycloheximide (CHX), we observed rapid loss of EMDΔ95–99-GFP within 4–8 hr (*Figure 3A–B*) while wild type EMD remained stable (*Figure 3—figure supplement 1*). This loss is blunted by co-treatment with the proteasome inhibitor MG132 (*Figure 3A*, third panel), consistent with our observations that INM protein turnover is slowed by proteasome inhibition using the RITE system (*Figure 2E–G*). If EMDΔ95–99-GFP is directed to the proteasome through ERAD, inhibition of earlier steps in this pathway should similarly cause accumulation of EMDΔ95–99-GFP. Indeed, pharmacological inhibition of p97 with the drug eeyarestatin I (*Wang et al., 2008*) causes modest accumulation of EMDΔ95–99 and of higher molecular weight species, a similar effect to proteasome inhibition itself (*Figure 3B*). In contrast, the drug kifunensine, which prevents ERAD targeting of misfolded glycosylated proteins (*Fagioli and Sitia, 2001*), has no effect on EMDΔ95–99-GFP levels as would be expected given the lack of glycosylation sites within EMD's small luminal domain (*Figure 3B*).

E3 ubiquitin ligases transfer ubiquitin to ERAD substrates, and each E3 ligase exhibits preference for a small number of substrates. A few E3 ligases have been implicated in ERAD of INM-localized substrates in yeast, including Doa10 and Asi1 (*Khmelinskii et al., 2014*). MARCH6 is a mammalian ortholog of Doa10 (*Zattas et al., 2016*). Mammalian orthologs of Asi1 have not been identified. Based on iterative sequence homology analysis through the MetaPhORs database (*Pryszcz et al., 2011*) we identified two possible Asi1 homologs: Rnf26 and CGRRF1. We depleted MARCH6, Rnf26, and CGRRF1 with short interfering RNA (siRNA), but observed no effect on EMDΔ95–99 protein levels (*Figure 3—figure supplement 2*), suggesting that these ligases do not catalyze EMD turnover, or alternatively that multiple E3 ligases are redundant in this process. Importantly, the broad group of ERAD-implicated E3 ligases rely on a handful of E2 ubiquitin conjugating enzymes for ubiquitin transfer. These E2 ligases – four in mammals – thus represent a key control point for ERAD (*Christianson et al., 2011*; *Leto et al., 2019*). We targeted these four E2 ubiquitin ligases by siRNA transfection (*Figure 3—figure supplement 2*) and a subset of these by inducible RNAi using a potent microRNA-based system (*Fellmann et al., 2013*) and analyzed the effects on EMDΔ95–99-GFP levels. To our surprise, knockdown of UBE2G1, UBE2G2, UBE2J1, and UBE2J2 either did not stabilize or instead decreased EMDΔ95–99-GFP levels (*Figure 3C*; *Figure 3—figure supplement 2*). This finding suggests that when ERAD is perturbed, EMD variants can be cleared by an alternative pathway.

## Emerin is subject to rapid stress-dependent clearance from the ER and NE

Global inhibition of ERAD places profound protein folding stress on the ER membrane network and induces the unfolded protein response (UPR) (*Christianson et al., 2011*). We considered whether direct induction of ER stress was sufficient to accelerate the turnover of EMDΔ95–99-GFP. We tested the effect of the ER stressor thapsigargin (THG), which disrupts ER homeostasis by causing release of Ca2+ from the ER lumen, on EMDΔ95–99 protein stability. Compared to CHX treatment alone (*Figure 3—figure supplement 1*), THG co-treatment further destabilized EMDΔ95–99-GFP (*Figure 3D*). Strikingly, when we tracked EMDΔ95–99 protein localization by time-lapse microscopy (*Figure 3E*), it became apparent that NE localization of EMDΔ95–99 significantly decreases within 2 hr of THG treatment, concomitant with accumulation in a perinuclear membrane compartment that morphologically resembles the Golgi apparatus. By 8 hr after THG administration, EMDΔ95–99 was undetectable (*Figure 3E–F*). These data suggest that under conditions of ER stress, EMDΔ95–99 is cleared from the NE/ER membrane network by transport out of the ER and eventual disposal of the protein in a post-ER compartment.

## Acute ER stress reroutes emerin through the secretory pathway

We took several approaches to test the possibility that EMDΔ95–99 leaves the NE/ER network during ER stress. Firstly, we made use of the characteristic sugar modifications that occur as cargoes progress through the secretory pathway to determine whether EMDΔ95–99 accesses post-ER compartments. Since the short lumenal domain of EMD lacks a glycosylation consensus site, we

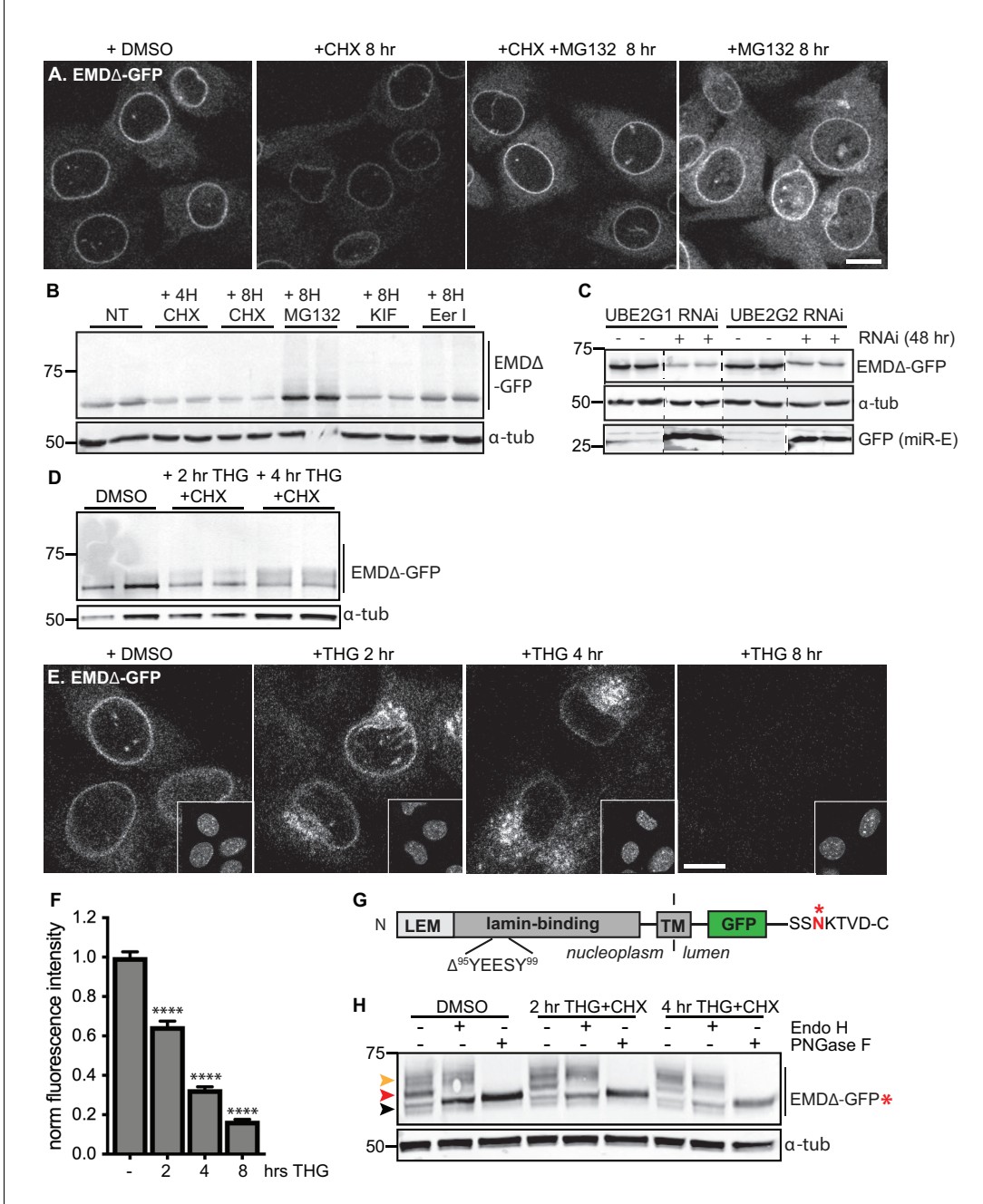

**Figure 3.** Acute stressors destabilize mutant emerin protein levels. (**A**) C2C12 cells stably expressing EMDΔ-GFP and treated with DMSO vehicle control, CHX alone, CHX and MG132, or MG132 alone for 8 hours. All images were acquired using the same laser power and detector gain settings. Single confocal z slices shown. (**B**) Western blot analysis of protein levels in C2C12 cells stably expressing EMDΔ-GFP and treated with DMSO vehicle, the translation inhibitor CHX, the proteasome inhibitor MG132, the p97 ATPase inhibitor eeyarestatin, or the glycosylation trimming inhibitor kifunensine for the time periods shown. a-tubulin shown as loading control. (**C**) Western blot analysis of U2OS cells stably expressing EMDΔ-GFP and doxycycline-inducible RNAi targeting the E2 ubiquitin ligases UBE2G1 and UBE2G2 and treated with DMSO vehicle control (-) or with doxycycline (+) for 48 hours. Free GFP indicates RNAi induction. a-tubulin shown as loading control. (**D**) Western blot detection of EMDΔ-GFP levels in cells treated with DMSO vehicle, or co-treated with CHX and the ER stress inducer THG for the time periods shown. a-tubulin shown as loading control. (**E**) C2C12 cells stably expressing EMDΔ-GFP and treated with vehicle control or with THG for the time periods shown. Insets show nuclei in the same ~50 μm field of view stained with Hoechst. All images acquired using the same laser power and detector gain settings; single confocal z slices shown. (**F**) Quantification of total NE-localized GFP fluorescence in maximum intensity projections of confocal z slices acquired across the conditions shown in (**E**) for N > 410 cells per condition. (**G**) Diagram of emerin domain organization and the sequence of an inserted C-terminal glycosylation sequence derived from the opsin protein, with glycosylation acceptor site marked (*). (**H**) Analysis of EMDΔ-GFP* glycosylation state in cells subjected to treatment with

*Figure 3 continued on next page*

*Figure 3 continued*

DMSO vehicle control or CHX and THG cotreatments for the times indicated. Red arrowhead indicates EndoH-sensitive glycosylated state of EMDΔ-GFP*; orange arrowhead indicates EndoH-resistant states of EMDΔ-GFP*; black arrowhead indicates deglycosylated EMDΔ-GFP*. a-tubulin shown as loading control. Numbers to left of blots indicate molecular weights in kDa. Scale bars in micrographs indicate 10 mm. See also Figure 3 – figure supplement 1, 2, and 3.

DOI: https://doi.org/10.7554/eLife.49796.006

The following figure supplements are available for figure 3:

**Figure supplement 1.** Localization and stability of a disease-linked emerin variant.

DOI: https://doi.org/10.7554/eLife.49796.007

**Figure supplement 2.** siRNA-mediated E2 or E3 ubiquitin ligase knockdowns do not stabilize EMDΔ-GFP.

DOI: https://doi.org/10.7554/eLife.49796.008

**Figure supplement 3.** Glycosylation reporter variants are destabilized by ER stress and recovered by BFA treatment.

DOI: https://doi.org/10.7554/eLife.49796.009

engineered the glycosylation site from the opsin protein (*Bulbarelli et al., 2002*) (SS**N**KTVD) onto the lumen-facing C terminus of EMDΔ95–99. If EMDΔ95–99 is retained in the ER, all of its N-linked glycans should remain sensitive to the trimming enzyme Endo H. On the other hand, if EMDΔ95–99 exits the ER, its N-linked glycans will be elaborated with further modifications so that Endo H can no longer trim them. These glycosylation states can be detected as progressive increases in molecular weight, and can be completely removed by incubation with the enzyme PNGase F. This engineered variant, EMDΔ95–99-GFP*, localizes normally to the NE and also disappears from the NE upon ER stress induction (*Figure 3—figure supplement 3*). In unstressed cells, EMDΔ95–99-GFP* is predominantly observed in an Endo H-sensitive glycosylation state (*Figure 3H*, red arrowhead), with a minor pool of Endo H-resistant protein (*Figure 3H*, orange arrowhead). In contrast, EMDΔ95–99-GFP* shifts progressively to a higher molecular weight, Endo H-resistant state over 2 to 4 hr of co-treatment with CHX and THG. This indicates that ER stress induction increases the proportion of EMDΔ95–99 that exits the NE/ER and samples post-ER compartments.

Upon ER stress induction, EMDΔ95–99-GFP accumulates in a perinuclear domain that resembles the Golgi apparatus (*Figure 3E*). We evaluated the extent of colocalization of EMDΔ95–99-GFP with the medial Golgi resident protein giantin in untreated cells and cells that had been treated with THG for 2–4 hr (*Figure 4A–C*). In untreated cells, EMDΔ95–99-GFP was not detectable in the Golgi, but THG treatment rapidly induced Golgi colocalization as NE-localized EMDΔ95–99 levels decreased (*Figure 4A–C*). Comparing GFP fluorescence intensity in the Golgi *versus* the NE over time revealed that ER stress induces significant enrichment of EMDΔ95–99-GFP in the Golgi accompanied by loss from the NE within 2 hr (*Figure 4D*).

EMDΔ95–99 could be delivered to the Golgi by vesicular transport from the ER. Transport between the ER and the Golgi is mediated by packaging of cargoes into COP-coated vesicles (*Barlowe and Miller, 2013*), a process which can be inhibited by the drug brefeldin A (BFA). BFA acts by disrupting COPI vesicle formation, leading to the collapse of the Golgi into the ER membrane network (*Chardin and McCormick, 1999*). To test whether clearance of EMDΔ95–99-GFP from the NE/ER requires vesicle-mediated ER-to-Golgi transport, we co-incubated cells expressing EMDΔ95–99-GFP with THG and BFA. Strikingly, co-treatment with BFA nearly quantitatively reversed loss of EMDΔ95–99-GFP from the NE (*Figure 4E–F*). Taken together with the time-dependent enrichment of EMD variants in the Golgi apparatus (*Figure 4A–C*), and the time-dependent accumulation of more complex N-glycosylated variants of EMDΔ95–99-GFP* (*Figure 3H*), this indicates that under ER stress, EMD variants can be cleared from both the NE and ER by vesicular transport through the Golgi.

While vesicle-mediated transport is the major pathway by which proteins move from the ER and onward through the secretory pathway, alternative modes of removing protein from the ER exist, in particular during ER stress. Recent evidence indicates that the ER can undergo autophagy under various conditions, including acute ER stress (*Smith et al., 2018*). To evaluate the possibility that EMDΔ95–99-GFP could be engulfed and removed from the NE and ER by autophagosomes, we tested the ability of the PI3K inhibitor KU55933 to reverse EMDΔ95–99-GFP loss. PI3K signaling promotes the formation of isolation membranes that engulf autophagic cargo (*Farkas et al., 2011*; *Klionsky et al., 2016*). We observed that cotreatment with KU55933 during acute ER stress could

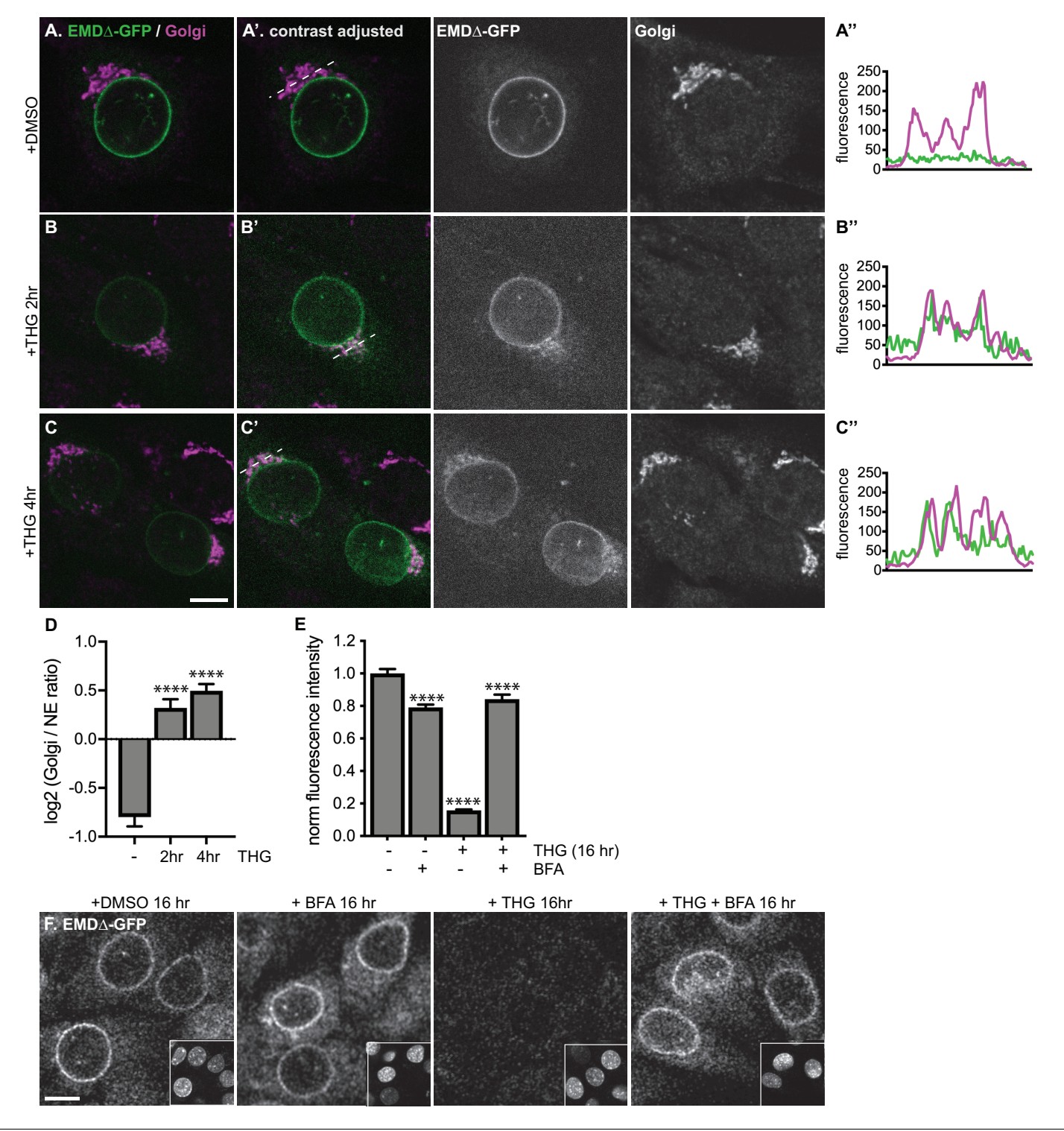

**Figure 4.** Stress-induced clearance of mutant emerin from the ER and NE involves the secretory pathway. (A-C) Representative confocal slices of cells stably expressing EMDΔ-GFP, treated with DMSO or THG for the indicated times and costained for giantin to mark the Golgi (magenta). All images were acquired using the same laser power and detector gain settings. (A'-C') Are contrast-adjusted to show relative levels of EMDΔ-GFP in NE and Golgi. Dotted lines mark positions of linescans in (A''-C''). (D) Quantification of GFP fluorescence intensity abundance ratio in Golgi *versus* NE in single, non-contrast-adjusted z slices over THG treatment timecourse. Columns indicate average with error bars indicating SEM for N > 37 cells from two independent experiments. **** indicates p-value<0.0001 compared to untreated (by t-test). (E-F) Representative confocal slices of cells stably expressing EMDΔ-GFP (F) after 16 hr of treatment with DMSO vehicle control, THG, BFA, or co-treatment with THG and BFA. All images were acquired

*Figure 4 continued on next page*

*Figure 4 continued*

using the same laser power and detector gain settings. Insets show nuclei in the same ~ 50 μm field of view stained with Hoechst. (E) Quantification of GFP fluorescence intensity at the NE in maximum intensity projections of confocal z series acquired across the conditions represented in (F). Columns indicate average with error bars indicating SEM for N > 386 cells from three independent experiments. **** indicates p-value<0.0001 compared to untreated (by t-test). Scale bars in micrographs indicate 10 μm. See also *Figure 4—figure supplement 1*.
DOI: https://doi.org/10.7554/eLife.49796.010
The following figure supplement is available for figure 4:

**Figure supplement 1.** Colocalization of emerin with the Golgi.
DOI: https://doi.org/10.7554/eLife.49796.011

not prevent loss of EMDΔ95–99-GFP from the NE/ER (*Figure 5A–C*), in contrast to the ability of BFA treatment to rescue EMD loss. This indicates that vesicle-mediated transport to the Golgi and not autophagic engulfment mediates EMDΔ95–99-GFP's exit from the ER during stress.

Proteins in post-ER compartments can be degraded by vesicle-mediated traffic to the lysosome (*Saftig and Klumperman, 2009*). To investigate whether the lysosome was the eventual destination of EMDΔ95–99-GFP after ER export, we co-incubated cells expressing EMDΔ95–99-GFP with THG and bafilomycin A1 (Baf A1), which impairs lysosome acidification and thus slows protein degradative processes within lysosomes.

Under these conditions, we observed complete translocation of EMDΔ95–99-GFP out of the NE and ER and into numerous vesicles that are decorated with the lysosomal marker LAMP1 (*Figure 5F*). Notably, bafilomycin A1 alone did not trap EMDΔ95–99-GFP in lysosomes (*Figure 5E*), indicating that ER stress potentiates exit from the NE/ER and lysosomal accumulation. This indicates that the eventual destination of EMDΔ95–99-GFP after export from the NE/ER network is the lysosome.

If EMDΔ95–99 arrives at the lysosome by trafficking through the secretory pathway, this should be accompanied by the accumulation of Endo H-resistant N-glycosylation modifications on our engineered reporter EMDΔ95–99-GFP*. Indeed, we observe that the majority of EMDΔ95–99-GFP* exists as an Endo H-sensitive species in unperturbed cells (*Figure 5G*, red arrowhead), but shifts progressively to a higher molecular weight, Endo H-resistant state over 2 to 4 hr of co-treatment with THG and Baf A1 (*Figure 5G*, orange arrowhead). This indicates that EMDΔ95–99 traverses the secretory pathway before being delivered to the lysosome.

## Emerin transiently accesses the plasma membrane during ER stress

A possible route from the early secretory pathway to the lysosome could involve anterograde transport following the 'bulk flow' of the secretory pathway, through the Golgi and into vesicles destined for the plasma membrane (PM). There, mislocalized proteins may be selectively endocytosed and trafficked to lysosomes for degradation through retrograde transport (*Saftig and Klumperman, 2009*). To explore this possibility, we performed antibody uptake assays in cells expressing EMDΔ95–99-GFP under homeostatic or stressed conditions. EMD is a tail-anchored protein with its final C-terminal amino acids facing the ER lumen; the C-terminal GFP tag will thus face the extracellular space if EMDΔ95–99-GFP accesses the PM (*Figure 6A*). We tested whether EMDΔ95–99's GFP tag is accessible to an anti-GFP antibody applied to the surface of intact cells. In untreated cells, a small amount of EMDΔ95–99-GFP (*Figure 6B–D*) is accessible to an anti-GFP antibody, but not to an anti-myc antibody, indicating that EMDΔ95–99 is not completely restricted to intracellular membrane compartments under homeostatic conditions. Importantly, the signal from the anti-GFP antibody is specific to cells that express a GFP fusion protein (*Figure 6D–E*). Upon induction of ER stress by THG, the amount of PM-accessible EMDΔ95–99-GFP rapidly increases within 2–4 hr, and begins to taper off within 6 hr. This implies that ER stress induces the export of EMDΔ95–99-GFP from the NE/ER to the PM as well as its internalization. Importantly, the GFP antibody signal is visible within intracellular puncta, consistent with EMDΔ95–99-GFP:antibody conjugates being rapidly internalized into vesicles after PM delivery. Based on the timescale when the levels of EMDΔ95–99-GFP begin to significantly decrease at the NE (*Figure 3E–F*), increase in the Golgi (*Figure 4A–D*), transit through the PM (*Figure 6*) and accumulate in lysosomes (*Figure 5F*), we infer that EMDΔ95–99-GFP is transported through the secretory pathway to the PM, then internalized and delivered to lysosomes for degradation.

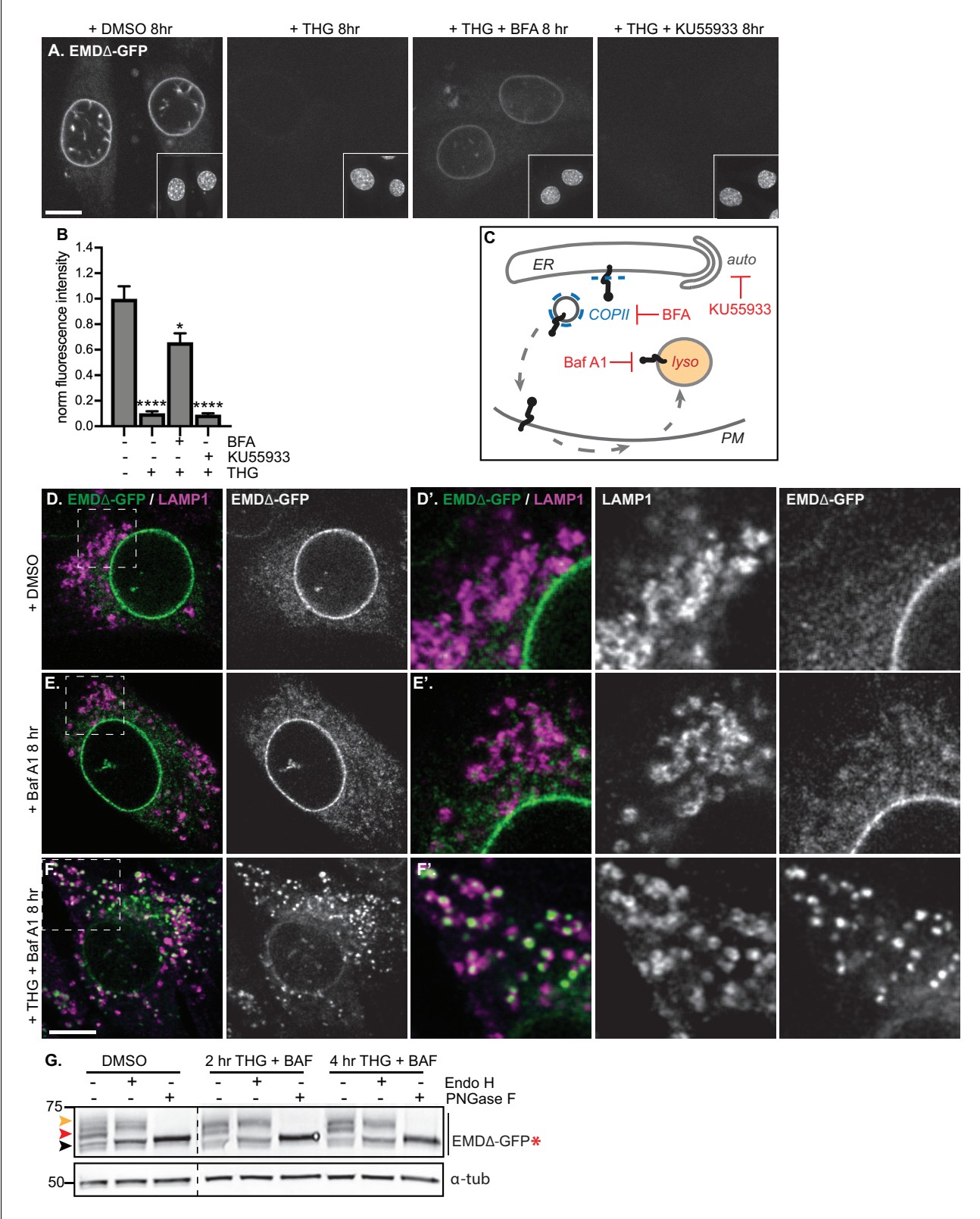

**Figure 5.** Mutant emerin trafficking is dependent on lysosomal but not autophagosomal function. (**A**) Representative confocal slices of cells stably expressing EMDΔ-GFP after 8 hr of treatment with DMSO vehicle control, THG, co-treatment with THG and BFA, or co-treatment with THG and KU55933. Insets show nuclei in the same field of view stained with Hoechst. (**B**) Quantification of NE-localized GFP fluorescence intensity in maximum intensity projections of confocal z series acquired across the conditions shown in (**A**). Columns indicate average and error bars indicate SEM for N > 56

*Figure 5 continued on next page*

Figure 5 continued
cells from three independent experiments. **** indicates p-value<0.0001 compared to untreated (by t-test). (C) Diagram of processes perturbed by KU55933, BFA, and Baf A1 treatment. (D-F) Representative confocal slices of C2C12 cells stably expressing EMDΔ-GFP and costained for LAMP1 after treatment with DMSO vehicle control (D), Baf A1 (E), or co-treatment with THG and Baf A1 (F) for the indicated times. (D'-F') Insets show GFP-LAMP1 colocalization within ~ 15 μm field of view demarcated by dashed rectangles in (D-F). (G) Analysis of EMDΔ-GFP* glycosylation state in cells subjected to treatment with DMSO vehicle control or THG and Baf A1 cotreatments for the times indicated. Red arrowhead indicates EndoH-sensitive glycosylated state of EMDΔ-GFP*; orange arrowhead indicates EndoH-resistant states of EMDΔ-GFP*; black arrowhead indicates deglycosylated EMDΔ-GFP*. α-tubulin shown as loading control. Numbers to left of blots indicate molecular weights in kDa. Scale bars in micrographs indicate 10 μm.
DOI: https://doi.org/10.7554/eLife.49796.012

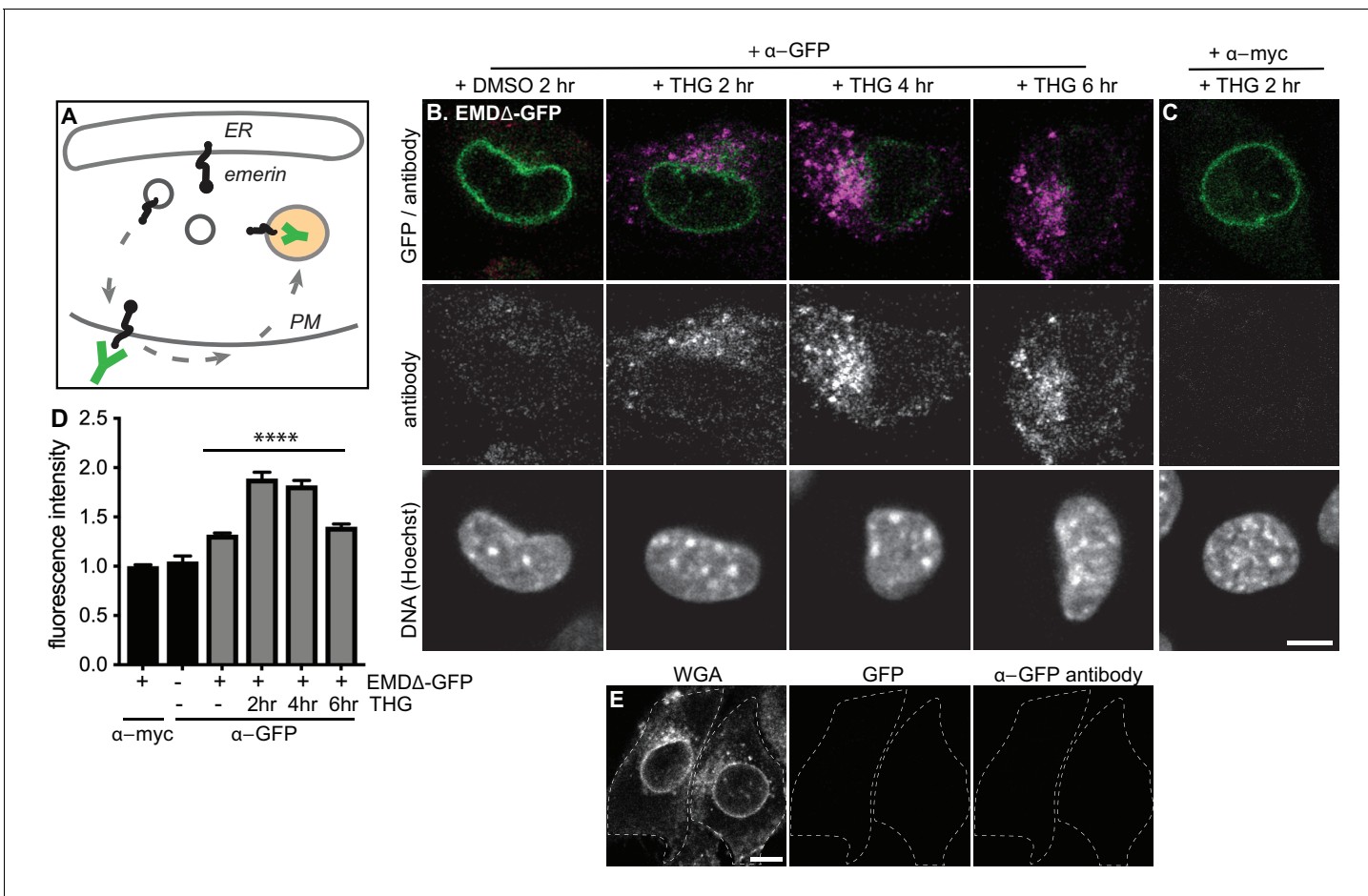

**Figure 6.** Mutant emerin traffics through the plasma membrane upon ER stress. (A) Schematic of antibody uptake assay experimental design. If emerin accesses the plasma membrane (PM), it will be detected by anti-GFP antibody (green), which will bind the surface-exposed GFP tag. (B) Uptake of anti-GFP antibody (magenta) by cells stably expressing EMDΔ-GFP and treated with DMSO vehicle control or THG for 2, 4, or 6 hours. Cells were incubated with anti-GFP antibody for the final hour of these treatment periods before fixation and imaging. (C) Control demonstrating lack of uptake of anti-myc antibody by cells stably expressing EMDΔ-GFP and treated with THG for 2 hours. (D) Quantification of internalized antibody signal in EMDΔ-GFP expressing cells. Columns indicate average and error bars indicate SEM for N > 235 cells from 3 independent experiments. **** indicates p-value < 0.0001 (t-test) compared to untreated. (E) Control demonstrating lack of uptake of anti-GFP antibody by untreated C2C12 cells that do not express a GFP fusion protein. WGA is used to define cell boundaries. All images were acquired using the same laser power and detector gain settings. Scale bars in micrographs indicate 10 μm. See also *Figure 6—figure supplement 1*.
DOI: https://doi.org/10.7554/eLife.49796.013
The following figure supplement is available for figure 6:

**Figure supplement 1.** Access of emerin to the plasma membrane.
DOI: https://doi.org/10.7554/eLife.49796.014

## Other INM proteins do not undergo stress-dependent clearance

Our findings indicate that EMDΔ95–99 is subject to proteasome-dependent turnover at the INM (*Figure 2G–H*; *Figure 3A–B*), but can also be rapidly removed from the INM and ER membrane network and targeted for degradation during ER stress. This raises the possibility that additional INM proteins are susceptible to stress-dependent degradation. To address this, we tested the response of additional INM proteins to ER stress induction by THG, to ER export blockage by BFA, and to lysosomal maturation blockage by Baf A1. We selected proteins with distinct topologies and half-lives, including the long-lived polytopic INM protein nurim and the less stable single-pass transmembrane protein Sun2 (see *Figure 1H*, Table S5). Prolonged treatment with THG modestly decreased nurim protein levels and significantly decreased Sun2 protein levels (*Figure 7A–B,D*), likely as a consequence of translational inhibition caused by ER stress (*Harding et al., 1999*). Consistent with this interpretation, the sensitivity of these two proteins tracks with the relative differences in their half-lives (Table S5); nurim has a half-life of ~ 9 days, while Sun2 has a half-life of ~ 3 days in non-dividing cells. Importantly, however, co-incubation with THG and BFA had no effect on either the subcellular localization or abundance of nurim or Sun2 (*Figure 7A–B,D*), indicating that loss of these proteins is not mediated by ER export. Further, neither protein leaves the NE/ER to accumulate in lysosomes when lysosome acidification is blocked by Baf A1 (*Figure 7A', B'*).

We next asked whether wild type EMD was also subject to this pathway. Similarly to EMDΔ95–99, NE-localized EMD-GFP decreases when stress is induced by THG, but remains stable when stress induction by THG is accompanied by secretory pathway disruption by BFA (*Figure 7C*). ER stress also induces EMD-GFP to enrich in the Golgi (*Figure 4—figure supplement 1*) and access the plasma membrane (*Figure 6—figure supplement 1*). EMD-GFP also accumulates in perinuclear puncta that are likely lysosomes when cells are co-incubated with THG and Baf A1 (*Figure 7C'*). We again engineered an opsin glycosylation site onto the C-terminus of EMD-GFP in order to track movement of EMD through membrane compartments. EMD-WT-GFP* localizes to the NE and responds similarly to ER stress and secretory pathway blockage (*Figure 7—figure supplement 1*). As we observed with EMDΔ95–99, EMD-WT-GFP* exists predominantly in an Endo H-sensitive modification state in unstressed cells (*Figure 7E–F*, red arrowheads). Higher molecular weight, Endo H-resistant species increase in abundance when cells are coincubated with THG and CHX (*Figure 7E*, orange arrowhead) or with THG and Baf A1 (*Figure 7F*, orange arrowhead). We thus conclude that wild type EMD is subject to the same stress-induced lysosomal degradation pathway as EMDΔ95–99. However, when EMD-GFP's response to THG is tracked over time, it is clear that displacement of wild type EMD from the NE proceeds significantly more slowly than displacement of EMDΔ95–99 (*Figure 8B,D*). This indicates that stress-dependent trafficking out of the INM is selective to variants of EMD, and that intrinsic features of EMD control its clearance from the NE/ER and targeting into lysosomes.

## A signal within emerin's LEM domain is required for stress-dependent export

Why are EMD variants selectively targeted for stress-dependent clearance from the INM and ER? We considered functional domains that might be involved in responding to ER stress. EMD is a tail-anchored protein with a ~ 10 amino acid tail that protrudes into the ER lumen (*Figure 3A*). This short sequence lacks any known motifs for engaging with proteins within the ER lumen. EMD's nucleoplasmic domain includes an N-terminal LEM domain and an internal lamin A-binding region (*Figure 3A*). Emerin relies on lamin A for targeting to the INM (*Vaughan et al., 2001*) but we noted that in *lmna* -/- MEFs, EMD-GFP remained stably expressed even while mislocalized to the peripheral ER (*Figure 8—figure supplement 2*). The small deletion within EMDΔ95–99 falls within the lamin A-binding domain, but does not appear to affect the protein's affinity for the lamina as judged by FRAP (*Figure 2—figure supplement 1*), even though this variant responds more potently than wild type EMD to ER stress (*Figure 8D*). Taken together, these observations indicate that dissociation from the lamina is not sufficient to promote clearance of EMD from the NE/ER membrane system.

We next evaluated whether interactions mediated by the LEM domain could contribute to stress-dependent EMD export. The LEM domain (*Figure 3A*) is a protein fold that binds with high affinity to the soluble nucleoplasmic protein BAF (*Lee et al., 2001*). We deleted this domain and queried the effects on EMD localization and trafficking. When expressed within unperturbed cells, EMDΔLEM

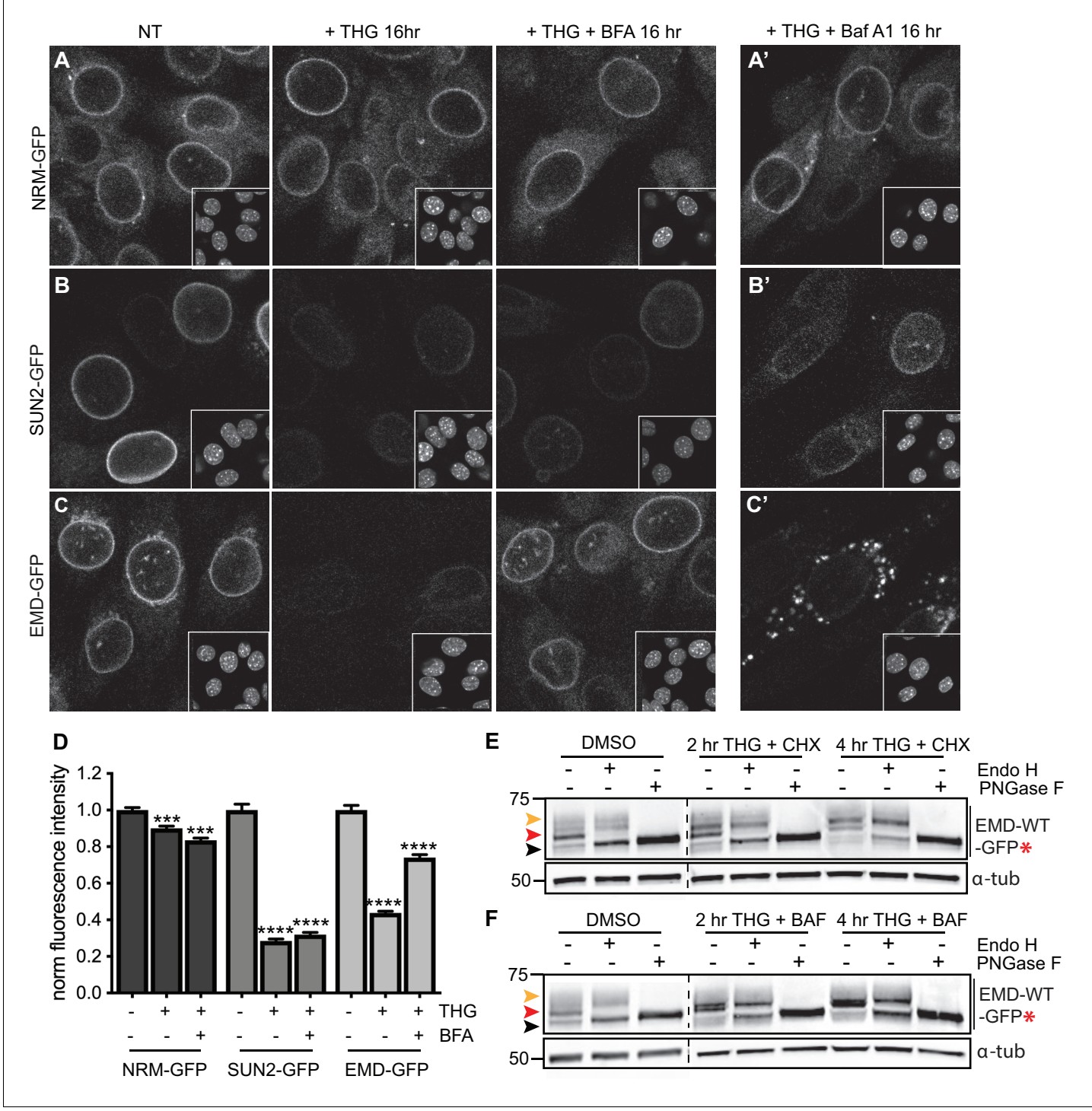

**Figure 7.** Emerin, but not other INM proteins, undergoes stress-dependent clearance from the NE and ER and accesses lysosomes. (A-C) Representative confocal slices of cells stably expressing NRM-GFP (A), Sun2-GFP (B), or EMD-GFP (C) after 16 hr of treatment with DMSO vehicle control, THG, or co-treatment with THG and BFA. Insets show nuclei in the same ~ 50 μm field of view stained with Hoechst. (A'-C') Representative confocal slices of cells co-treated with THG and Baf A1. All images were acquired using the same laser power and detector gain settings. (D) Quantification of GFP fluorescence intensity at the NE in maximum intensity projections of confocal z series acquired across conditions represented in (A-C). Columns indicate average and error bars indicate SEM for N > 690 cells from three independent experiments. **** indicates p-value<0.0001 compared to untreated (t-test). (E-F) Analysis of EMD-WT-GFP* glycosylation state in cells subjected to treatment with DMSO vehicle control or THG and CHX (E) or THG and Baf A1 (F) cotreatments for the times indicated. Red arrowhead indicates EndoH-sensitive glycosylated state of EMD-WT-GFP*; orange arrowhead indicates EndoH-resistant states of EMD-WT-GFP*; black arrowhead indicates deglycosylated EMD-WT-GFP*. α-tubulin

*Figure 7 continued on next page*

*Figure 7 continued*

shown as loading control. Numbers to left of blots indicate molecular weights in kDa. Scale bars in micrographs indicate 10 μm. See also *Figure 7—figure supplement 1*.

DOI: https://doi.org/10.7554/eLife.49796.015

The following figure supplement is available for figure 7:

**Figure supplement 1.** Glycosylation reporter variants are destabilized by ER stress and recovered by BFA treatment.

DOI: https://doi.org/10.7554/eLife.49796.016

exhibited normal enrichment in the NE (*Figure 8C*), consistent with its ability to bind the lamina independently of the LEM domain. However, we observed that this mutant was less responsive than other EMD variants to ER stress induction; NE-localized EMDΔLEM was clearly detectable over several hours of THG treatment and was significantly less sensitive than full-length EMD to ER stress (*Figure 8C–E*). We surmise that the eventual loss of EMDΔLEM results from translational inhibition resulting from ER stress (*Harding et al., 1999*) and degradation by alternative pathways. Consistent with the interpretation that the LEM domain mediates post-ER trafficking, co-incubation of EMDΔLEM-expressing cells with THG and BFA or Baf A1 each had no effect on protein levels or localization (*Figure 8F–H*). These results suggest that without the LEM domain, EMD does not access post-ER compartments. To directly evaluate this, we generated a glycosylation-reporting variant, EMDΔLEM-GFP*, with a glycosylation consensus site at the lumenal C terminus. In contrast to variants of EMD with an intact LEM domain, EMDΔLEM-GFP* accumulates only Endo H-sensitive modifications and remains equivalently Endo H-sensitive during ER stress and lysosome blockage (*Figure 8I–J*, red arrowheads). Altogether, these data indicate that a signal within emerin's LEM domain enables selective export from the ER under stress conditions.

## Discussion

In this work, we applied a dynamic proteomic strategy to define organelle-wide trends in protein turnover across the NE/ER membrane network in mammalian cells. While the INM is separated from the bulk ER by the selective barrier of the NPC, we observe no difference in global protein turnover rates between the ER and INM compartments or any correlation between INM protein size and turnover kinetics. This, along with specific visualization of mature INM proteins by RITE tagging and microscopy (*Figure 2*) and previous studies (*Tsai et al., 2016*), suggests that turnover of INM proteins can be effectively achieved in situ.

Moving forward with the rapidly turned over INM protein EMD as a model substrate for dissecting INM protein turnover, we identified an even less stable, EDMD-linked variant of EMD as an ideal substrate for sensitively probing INM protein turnover pathways. We noted that turnover of maturely folded EMD and EMDΔ95–99 exhibits proteasome dependence at the INM (*Figure 2F–H*; *Figure 3A*), while nascent EMDΔ95–99 accumulates in multiple cellular compartments when the proteasome is inhibited (*Figure 3A*, fourth panel). Taken together, these observations lead us to infer that mature EMD variants and potentially other INM proteins can be turned over in situ at the INM by a pathway that terminates in proteasomal degradation, while immature EMD variants (and potentially other INM proteins) are also subject to co-translational quality control that terminates in proteasomal degradation. As proteasomal inhibition and p97 inhibition each stabilize EMDΔ95–99 (*Figure 3B*), we expect that EMDΔ95–99 is an ERAD client under some conditions.

Surprisingly, however, we also find that EMD can be selectively shunted to an alternative turnover pathway under conditions of acute ER stress. This pathway is rapidly induced by ERAD blockage or by pharmacological induction of acute ER protein folding stress (*Figure 3*) and requires ER export (*Figures 3* and *4*). Notably, changes to EMDΔ95–99 localization and levels are apparent at a time-scale shorter than the normal half-life of EMDΔ95–99, within 2–4 hr of ER stress induction. Based on the transient appearance of EMDΔ95–99 at the PM (*Figure 6*) and its accumulation in lysosomes (*Figure 5*), we conclude that EMD transits through the secretory pathway and is then internalized into lysosomes. While our data indicate that a significant proportion of EMD leaves the NE/ER during ER stress, we cannot rule out the possibility that ERAD-mediated degradation of some proportion of EMD takes place within the NE/ER network in parallel to the lysosome-mediated pathway

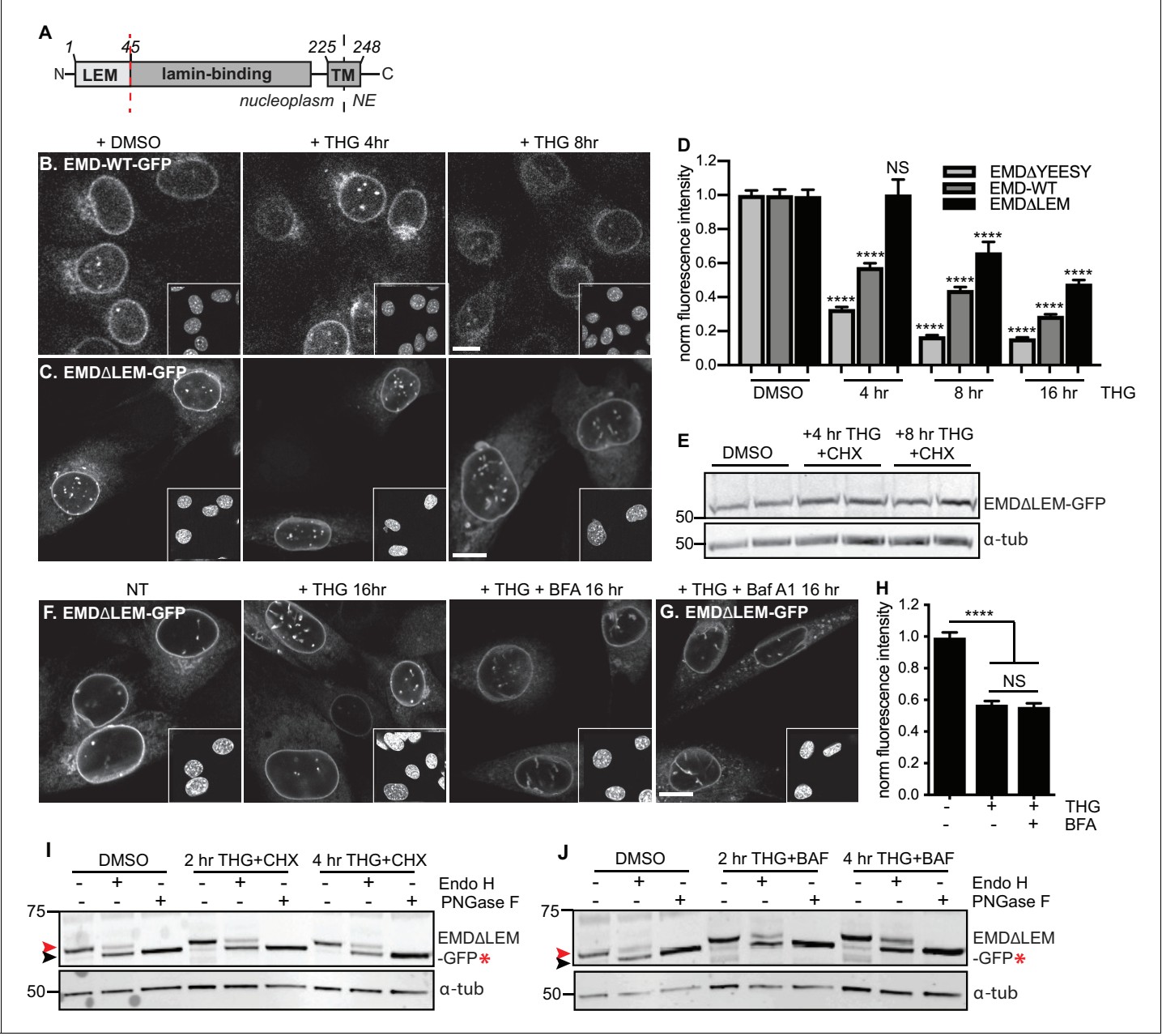

**Figure 8.** A signal within emerin's. LEM domain is required for stress-dependent clearance from the NE and ER (A) Diagram of emerin domain organization with N-terminal LEM domain deletion indicated (amino acids 1-45). (B-C) Representative confocal slices of C2C12 cells stably expressing EMD-WT-GFP (B) or EMDΔLEM-GFP (C) after treatment with DMSO vehicle control or THG for the indicated times. Insets show nuclei in the same ~50 μm field of view stained with Hoechst. (D) Quantification of EMDΔ95-99-GFP (as also shown in Figure 3F), EMD-WT-GFP, and EMDΔLEM-GFP fluorescence intensity at the NE in maximum intensity projections of confocal z series acquired across the timecourse shown in (B-C). Columns indicate average and error bars indicate SEM for N > 146 cells from 3 independent experiments. **** indicates p-value < 0.0001 compared to untreated (t-test). (E) Western blot of EMDΔLEM-GFP in cells treated with DMSO vehicle control or co-treated with THG and CHX for the indicated times. (F-G) Representative confocal slices of C2C12 cells stably expressing EMDΔLEM-GFP after treatment with (F) DMSO vehicle control, THG, THG + BFA, or (G) THG + Baf A1. (H) Quantification of GFP fluorescence intensity at the NE across the conditions shown in (F). Columns indicate average and error bars indicate SEM for N > 776 cells from 3 independent experiments. **** indicates p-value < 0.0001 compared to untreated (t-test). (I-J) Analysis of EMD-Δ LEM-GFP* glycosylation state in cells subjected to treatment with DMSO vehicle control or THG and CHX (I) or THG and Baf A1 (J) cotreatments for the times indicated. Red arrowhead indicates EndoH-sensitive glycosylated state of EMDΔLEM-GFP*; black arrowhead indicates deglycosylated EMDΔLEM-GFP*. a-tubulin shown as loading control. Numbers to left of blots indicate molecular weights in kDa. Scale bars in micrographs indicate 10 mm. See also *Figure 8—figure supplements 1* and *2*.

DOI: https://doi.org/10.7554/eLife.49796.017

*Figure 8 continued on next page*

*Figure 8 continued*

The following figure supplements are available for figure 8:

**Figure supplement 1.** Stability of EMDΔLEM-GFP over time of cycloheximide treatment.

DOI: https://doi.org/10.7554/eLife.49796.018

**Figure supplement 2.** Emerin is mislocalized to the ER, but not degraded in *Lmna* - /- MEFs.

DOI: https://doi.org/10.7554/eLife.49796.019

that we have identified. Nonetheless, this dynamic and selective removal of an INM protein is quite surprising and is inconsistent with models of the INM as a terminal depot for its resident proteins.

Our findings have some intriguing parallels to the fate of a misfolded variant of the GPI-anchored prion protein, PrP, during ER stress (*Satpute-Krishnan et al., 2014*). PrP is normally targeted to the PM, but a misfolded variant is retained within the ER by persistent association with protein folding chaperones. Similarly to what we observe for an INM protein, ER stress induces the rapid export of misfolded PrP through the secretory pathway, followed by transit through the PM and internalization and delivery to lysosomes for degradation. This mode of clearance has been referred to as rapid ER stress-induced export, or RESET (*Satpute-Krishnan et al., 2014*).

There are several notable contrasts between PrP's export from the peripheral ER and EMD's export from the INM and ER. For instance, the topologies of PrP and EMD are quite disparate. As a GPI-anchored protein, misfolded PrP faces the lumen of the ER, and an interaction between PrP and Tmp21, a sorting adaptor for luminal proteins, controls RESET (*Satpute-Krishnan et al., 2014*). Misfolded PrP remains associated with additional luminal ER-derived proteins during its transit through the secretory pathway, and these associations appear to enable recognition of misfolded PrP at the cell surface for internalization (*Zavodszky and Hegde, 2019*). In contrast, EMD is a tail-anchored protein, and interactions mediated by EMD's nucleoplasmic-facing LEM domain (*Figure 8*) control its stress-dependent clearance. We do not yet know whether EMD remains associated with other proteins as it transits through the secretory pathway, or what role those associations might play in targeting EMD for degradation.

PrP and EMD also exhibit distinct subcellular localization when not undergoing RESET. Misfolded PrP is retained in the ER network until RESET is initiated, while EMD is enriched in the INM and associated with the nuclear lamina. Importantly, EMD is small enough (~25 kDa) to diffuse freely across the NPC, meaning that it may release INM-localized binding partners and sample the ER with some frequency. This spatial separation between EMD's normal site of enrichment and its site of ER export may explain the longer timescale of RESET for EMD (2–4 hr) compared to ER-localized misfolded PrP (~1 hr) (*Satpute-Krishnan et al., 2014*).

Finally, PrP and EMD variants exhibit clear differences in selectivity for the RESET pathway. Only misfolded, ER-retained mutants of PrP are subject to RESET. On the other hand, both wild type EMD (*Figure 7*) and a less stable disease-linked variant (EMDΔ95–99) are subject to stress-dependent clearance, although EMDΔ95–99 is more rapidly cleared from the NE and ER. Both EMD variants appear functional until ER stress is induced, as judged by their localization and affinity for the INM (*Figure 3*, *Figure 2—figure supplement 1*). This suggests that clearance of EMD from the NE/ER is not strictly contingent on protein misfolding.

We find that selective, stress dependent clearance of EMD depends on its 45-amino acid LEM domain. LEM domains bind dynamically to the small soluble protein BAF, which exists in both nuclear and cytoplasmic pools (*Shimi et al., 2004*). While glycosylation reporters indicate that EMD variants also exit the NE/ER with some frequency under homeostatic conditions (*Figure 3H*, *Figure 7E–F*), this is completely abolished by deletion of the LEM domain (*Figure 8*). One model that could explain the dichotomy between LEM-mediated BAF binding and LEM-mediated ER export is that BAF and ER export-promoting factor(s) bind competitively to the same surface of EMD's LEM domain (*Figure 9*). It could be that acute ER stress is relayed to EMD via a structural reorganization or post-translational modification that disrupts the LEM:BAF interface. It is possible that this system could be used to rapidly remove EMD in response to ER stress and potentially other physiological stressors. This could in turn rapidly inhibit the normal functions of EMD at the INM, including participating in mechanosensitive signaling pathways (*Guilluy et al., 2014*) and contributing to the scaffolding of heterochromatic domains at the nuclear periphery (*Demmerle et al., 2013*).

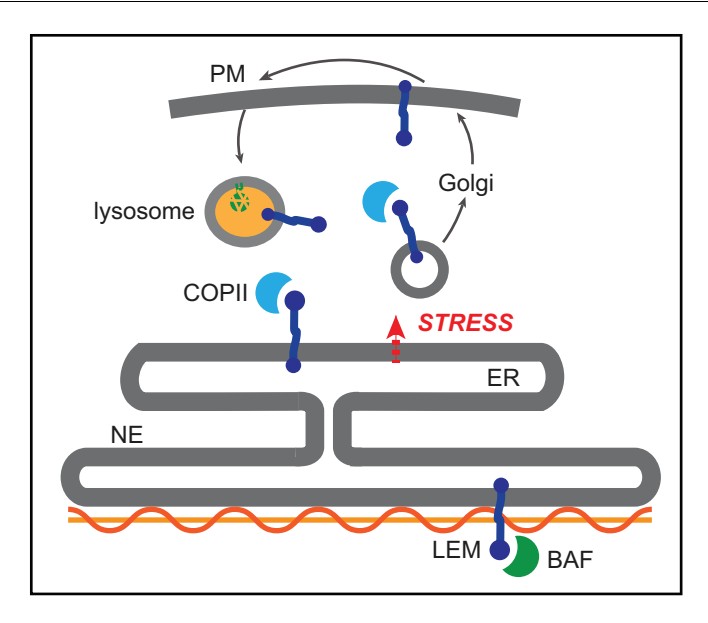

**Figure 9.** Competition model for emerin sorting via its LEM domain competitively binding to BAF or to the ER export machinery.

DOI: https://doi.org/10.7554/eLife.49796.020

Overall, our findings indicate that the INM can be rapidly remodeled in response to environmental stimuli, and that the function of the INM protein EMD may be dynamically controlled by integration of environmental inputs via its LEM domain.

Notably, muscular dystrophy and cardiomyopathy diseases are caused by loss-of-function mutations to EMD, many of which further destabilize the protein (*Bonne and Quijano-Roy, 2013*; *Fairley et al., 1999*). We find evidence that a muscular dystrophy-linked EMD variant (EMDΔ95–99) is more rapidly degraded under acute stress conditions, suggesting that an overzealous response to ER stress could contribute to the pathogenesis of EDMD. EMD is broadly expressed (*Uhlen et al., 2015*), but mutations predominantly affect muscle tissues. Intriguingly, skeletal muscle undergoes significant ER stress both during development and during normal function (*Deldicque et al., 2012*). We speculate that these features of muscle physiology may make muscle-localized EMD mutants especially vulnerable to ER stress-induced degradation.

## Materials and methods

**Key resources table**

| Reagent type (species) or resource | Designation | Source or reference | Identifiers | Additional information |
|---|---|---|---|---|
| Gene (*Mus musculus*) | emerin | | NCBI RefSeq NM_007927 | |
| Gene (*Mus musculus*) | nurim | | NCBI RefSeq NM_134122 | |
| Gene (*Mus musculus*) | Sun2 | | NCBI RefSeq NM_001205346 | |
| Cell line (*Mus musculus*) | C2C12 | ATCC | CRL-1772 | |

*Continued on next page*

*Continued*

| Reagent type (species) or resource | Designation | Source or reference | Identifiers | Additional information |
|---|---|---|---|---|
| Cell line (*Homo sapiens*) | U-2-OS | ATCC | HTB-96 | |
| Recombinant DNA reagent (plasmid) | pQCXIB vector | *Campeau et al. (2009)* Addgene | | Retroviral construct for stable expression |
| Recombinant DNA reagent (plasmid) | Myc/FLAG RITE vector | *Toyama et al. (2019)* | | Lentiviral contruct for stable expression of RITE-tagged protein |
| Recombinant DNA reagent (plasmid) | pQCXIB emerin-GFP | This paper | | Retroviral construct for stable expression |
| Recombinant DNA reagent (plasmid) | pQCXIB emerin-D95-99-GFP | This paper | | Retroviral construct for stable expression |
| Recombinant DNA reagent (plasmid) | pQCXIB emerin-DLEM-GFP | This paper | | Retroviral construct for stable expression |
| Recombinant DNA reagent (plasmid) | pQCXIB emerin-GFP-SSNKTVD | This paper | | Retroviral construct for stable expression |
| Recombinant DNA reagent (plasmid) | pQCXIB emerin-Δ95–99-GFP-SSNKTVD | This paper | | Retroviral construct for stable expression |
| Recombinant DNA reagent (plasmid) | pQCXIB emerin-ΔLEM-GFP-SSNKTVD | This paper | | Retroviral construct for stable expression |
| Recombinant DNA reagent (plasmid) | pQCXIB Sun2-GFP | This paper | | Retroviral construct for stable expression |
| Recombinant DNA reagent (plasmid) | pQCXIB nurim-GFP | This paper | | Retroviral construct for stable expression |
| Recombinant DNA reagent (plasmid) | Emerin-RITE | This paper | | Lentiviral contruct for stable expression of RITE-tagged protein |
| Recombinant DNA reagent (plasmid) | Nurim-RITE | This paper | | Lentiviral contruct for stable expression of RITE-tagged protein |

*Continued on next page*

*Continued*

| Reagent type (species) or resource | Designation | Source or reference | Identifiers | Additional information |
|---|---|---|---|---|
| Recombinant DNA reagent (plasmid) | Emerin-Δ95–99-RITE | This paper | | Lentiviral contruct for stable expression of RITE-tagged protein |
| Antibody | Rabbit polyclonal anti-emerin | Santa Cruz Biotechnology | Sc-15378 | WB (1:1000) |
| Antibody | GFP | Abcam | ab290 | Ab uptake (1:500); WB (1:1000) |
| Antibody | Mouse monoclonal anti-FLAG | Sigma-Aldrich | F1804 | IF (1:1000) |
| Antibody | Mouse monoclonal anti-Myc | Cell Signaling | 2233 | IF (1:1000); Ab uptake (1:500) |
| Antibody | Mouse monoclonal anti-tubulin | Sigma-Aldrich | T5168 | WB (1:2500) |
| Antibody | giantin | BioLegend | PRB-114C | IF (1:1000) |
| Antibody | LAMP1 | Abcam | ab24170 | IF(1:100) |
| Other | Alexa-647 WGA | Life Technologies | W32466 | IF (5 ug/ml) |
| Commercial assay or kit | PNGase F | NEB | P0704 | |
| Commercial assay or kit | Endo H | NEB | P0702 | |
| Chemical compound, drug | Thapsigargin | Thermo Fisher | T7459 | Used at 100 nM |
| Chemical compound, drug | MG132 | Cayman Chemical | 1211877-36-9 | Used at 10 uM |
| Chemical compound, drug | Bafilomycin A1 | BioViotica | BVT-0252 | Used at 100 nM |
| Chemical compound, drug | Brefeldin A | Tocris | 1231 | Used at 2.5 uM |
| Chemical compound, drug | Leupeptin | Sigma-Aldrich | L5793 | Used at 125 uM |
| Chemical compound, drug | cycloh eximide | Sigma-Aldrich | C-7698 | Used at 200 ug/ml |
| Other | $^{13}C_6$-Lysine | Cambridge Isotopes | CLM-2247 | |
| Other | $^{13}C_6$, $^{15}N_4$-Arginine | Cambridge Isotopes | CNLM-539 | |
| Other | Lysine/arginine free DMEM | Thermo Fisher | 88364 | |
| Other | Dialyzed fetal bovine serum | Thermo Fisher | 26400044 | |
| Other | Hoechst stain | Molecular Probes | H1399 | Used at 10 ug/ml |

*Continued on next page*

*Continued*

| Reagent type (species) or resource | Designation | Source or reference | Identifiers | Additional information |
|---|---|---|---|---|
| Recombinant DNA reagent (plasmid) | UBE2G1 miR-E LT3GEPIR | *Knott et al., 2014* | | TGCTGTTGACAGTG AGCGAAAGACAGC TGGCAGAACT CAATAGTGAAGCCA CAGATGTATTGA GTTCTGCCAGCT GTCTTCTGCC TACTGCCTCGGA |
| Recombinant DNA reagent (plasmid) | UBE2G2 miR-E LT3GEPIR | *Knott et al., 2014* | | TGCTGTTGACA GTGAGCGAACCG GGAGCAGTTCTATAAG ATAGTGAAGCCACAGA TGTATCTTATAGAA CTGCTCCCGGTCTG CCTACTGCCTCGGA |
| Recombinant DNA reagent (plasmid) | UBE2J1 miR-E LT3GEPIR | *Knott et al., 2014* | | TGCTGTTGACA GTGAGCGAAAGGTTG TCTACTTCA CCAGATAGTGA AGCCACAGATGTATC TGGTGAAGTAGACAACC TTCTGCCTA CTGCCTCGGA |
| Recombinant DNA reagent (plasmid) | MARCH6 miR-E LT3GEPIR | *Knott et al., 2014* | | TGCTGTTGACAGTG AGCGACTGGATC TTCATTCTTATTTA TAGTGAAGCCACA GATGTATAAATAAGA ATGAAGATCCA GCTGCCTAC TGCCTCGGA |
| Software, algorithm | Fiji | https://fiji.sc/ | | |
| Software, algorithm | RStudio | https: //rstudio.com/ | | |

## SILAC labeling

SILAC labeling was performed as a pulse-chase (*Ong and Mann, 2006*). Proliferating C2C12 mouse myoblasts were subcultured for > 5 population doublings in culture medium containing stable heavy isotopes of lysine and arginine ($^{13}C_6$-Lysine, $^{13}C_6$, $^{15}N_4$-Arginine) to completely label the cellular proteome. Cells were grown in SILAC-formulated DMEM lacking lysine and arginine and supplemented with 20% dialyzed FBS, penicillin/streptomycin, and SILAC amino acids. Complete label incorporation was verified by LC-MS/MS. Myoblasts were then grown to confluency and switched to differentiation medium containing heavy isotopes for 5 days to induce myotube differentiation. Differentiation medium contained SILAC DMEM, 2% dialyzed FBS, penicillin/streptomycin, and SILAC amino acids. Media was refreshed every other day. After differentiation, the mature myotube culture was switched to low serum differentiation medium containing the normal isotopes of lysine and arginine: $^{12}C_6$-Lysine, $^{12}C_6$, $^{14}N_4$-Arginine for 1–3 days.

Crude nuclear extracts were prepared similarly to previous work (*Buchwalter and Hetzer, 2017*; *Schirmer et al., 2003*). Cells were harvested by scraping into PBS, then swollen in hypotonic lysis buffer (10 mM potassium acetate, 20 mM Tris acetate pH 7.5, 0.5 mM DTT, 1.5 mM MgCl2, and protease inhibitors), followed by mechanical lysis through a 25-gauge needle and syringe. The nuclei were pelleted and the supernatant (containing cytosol) was decanted. Nuclei were then resuspended in buffer containing 10 mM Tris pH 8.0, 10% sucrose, 1 mM DTT, 0.1 mM MgCl2, 20 ug/ml DNase I, and 1 ug/ml RNase I. After nuclease treatment, nuclei were layered on top of a 30% sucrose cushion and pelleted. Crude nuclei were then extracted in 10 mM Tris pH 8, 1% n-octyl glucoside, 400 mM NaCl, and 1 mM DTT, and extracts and pellets were prepared separately for liquid chromatography-mass spectrometry.

## Lc-ms/MS

Samples were denatured in 8M urea/100 mM TEAB, pH 8.5; reduced with TCEP; alkylated with chloroacetamide; and digested overnight with trypsin. Digestion was quenched with 5% formic acid. Detergent was removed from pulse-labeled SILAC samples with SCX tips (EMD Millipore). Samples were run on a Thermo Orbitrap Fusion Tribrid MS/MS with CID fragmentation. The digest was injected directly onto a 30 cm, 75 um ID column packed with BEH 1.7 um C18 resin. Samples were separated at a flow rate of 200 nl/min on a nLC 1000. Buffer A and B were 0.1% formic acid in water and acetonitrile, respectively. A gradient of 1–25% B over 160 min, an increase to 35% B over 60 min, an increase to 90% B over another 10 min and a hold at 90%B for a final 10 min of washing was used for a total run time of 240 min. The column was re-equilibrated with 20 ul of buffer A prior to the injection of sample. Peptides were eluted directly from the tip of the column and nanosprayed into the mass spectrometer by application of 2.5 kV voltage at the back of the column. The Orbitrap Fusion was operated in data dependent mode. Full MS1 scans were collected in the Orbitrap at 120K resolution with a mass range of 400 to 1500 m/z and an AGC target of 4e. The cycle time was set to 3 s, and within this 3 s the most abundant ions per scan were selected for CID MS/MS in the ion trap with an AGC target of 1e and minimum intensity of 5000. Maximum fill times were set to 50 ms and 100 ms for MS and MS/MS scans, respectively. Quadrupole isolation at 1.6 m/z was used, monoisotopic precursor selection was enabled, charge states of 2–7 were selected and dynamic exclusion was used with an exclusion duration of 5 s.

## Analysis of proteomic data

Peptide and protein identification, quantification, and analysis were performed with Integrated Proteomics Pipeline (IP2) (Integrated Proteomics Applications; www.integratedproteomics.com). Tandem mass spectra were extracted from raw files using RawConverter (*He et al., 2015*) and searched with ProLUCID (*Xu et al., 2015*) against the mouse UniProt database (ID). The search space included all fully tryptic and half-tryptic peptide candidates. Carbamidomethylation on cysteine was allowed as a static modification. Data were searched with 50 ppm precursor ion tolerance and 600 ppm fragment ion tolerance. Data were filtered to 10 ppm precursor ion tolerance post-search. Identified proteins were filtered using DTASelect (*Tabb et al., 2002*) and utilizing a target-decoy database strategy to control the false discovery rate to 1% at the protein level.

Census2 (*Park et al., 2014*) was used for quantitative analysis of SILAC-labeled peptides. Peptides were subjected to stringent quality control criteria before inclusion in half-life determination analyses. Firstly, any peptide with a profile score < 0.8 was discarded. Secondly, peptides were filtered based on the extent of correlation between the heavy and light chromatograms, which is quantified as a regression score in Census. Peptides with extreme area ratios (less than 0.111 or greater than 9) were retained only if their regression score was > 0. Peptides with intermediate area ratios (between 0.111 and 9) were retained only if their regression score was > 0.8.

## Half-life calculation

For estimation of protein half-lives, we restricted our analysis to peptides that were detected in at least three timepoints. 1685 proteins passed this criterion with at least one peptide; individual peptides and protein-level data are reported in Tables S1 and S2, respectively. Area ratio values were transformed into % old values using the following equation:

$$\%\mathrm{old} = 100 * (1/(1 + \mathrm{AR}))$$

And individual peptides were fit to a line corresponding to the following equation:

$$\mathrm{Ln}(\%\mathrm{old}) = \mathrm{kt} + \mathrm{a}$$

Individual peptide fits with $r^2 > 0.8$ and values of k < 0 were retained for protein-level estimation of half-life. The slope of the fit for all peptides detected were averaged to produce an average value and standard deviation at the protein level. These average slope values were converted to half-life estimates using the equation below.

$$\mathrm{T}1/2 = -\ln(2)/\mathrm{k}$$

These values are reported in Table S3 for 1677 proteins. While calculated half-lives range from ~0.33 days to ~30 days, we note that half-lives at either extreme should be considered rough estimates of protein stability. For illustration, we have included example fits for proteins in Figure S1 with predicted half-lives of 0.5 day, 1 day, 2 days, 4 days, 8 days, and 16 days. Linear regression predicts half-life well under conditions where a line can be fitted with high fidelity and a non-zero slope is detectable. We note the good performance and clear distinctions in slope for proteins with predicted half-lives ranging from 1 to 8 days, and observed more frequent deviations in linearity at the low extreme (predicted T1/2 < 1 day) and slopes approaching zero at the high extreme (predicted T1/2 > 8 days). We expect that these factors limit the precision of half-life determination below 1 day and above 8 days from a 3 day timecourse. Shorter or longer timecourses would be required to investigate turnover at these timescales.

The TMHMM server (*Krogh et al., 2001*) was used to define the positions of transmembrane domains in INM proteins and infer extraluminal domain sequences.

## Plasmid construction

All emerin constructs are based on the emerin sequence from mouse (Uniprot ID O08579); all nurim constructs based on the nurim sequence from mouse (Uniprot ID Q8VC65); and all Sun2 constructs based on the Sun2 sequence from mouse (Uniprot ID Q8BJS4). Emerin, Sun2, and nurim were each C-terminally tagged with GFP by stitching PCR (*Heckman and Pease, 2007*), where each open reading frame was amplified with the start codon included and the stop codon omitted, while GFP was amplified with its start codon omitted and stop codon included. Primersets were designed with overhangs including homology between each ORF and GFP, so that a second round of PCR with flanking primers and the first two PCR products used as templates generates an ORF-GFP fusion. EMDΔ95–99 was generated by Quickchange mutagenesis of the emerin-GFP sequence followed by sequence verification. EMDΔLEM-GFP was constructed by stitching PCR of emerin nucleic acid sequence 136–777 corresponding to residues 46–258 of emerin protein with the stop codon omitted. A new ATG start codon was introduced by PCR, and the C-terminal GFP tag introduced by stitching PCR. All ORFs were introduced into the pQCXIB vector (*Campeau et al., 2009*) for retroviral delivery and constitutive expression under a CMV promoter by Gateway cloning.

INM-RITE tag plasmids were constructed as described in *Toyama et al. (2019)*. ORFs of interest were introduced into the FLAG/myc-RITE or myc/FLAG-RITE plasmid backbones, then the entire ORF-RITE construct was amplified and recombined into a pDONR207 Gateway entry vector, followed by recombination into the pQCXIB retroviral vector for constitutive mammalian expression.

Glycosylation reporter plasmids were constructed by introducing a 21 base-pair sequence encoding the glycosylation acceptor site SSNKTVD within a 3' PCR primer for amplification of EMD-WT-GFP, EMDΔYEESY-GFP, and EMDΔLEM-GFP. The resulting PCR product was inserted into the pQCXIB vector by Gateway cloning. Sequence verified clones were used for stable cell line generation.

UBE2G1, UBE2G2, UBE2J1, and MARCH6 miR-E inducible RNAi plasmids were constructed as described in *Fellmann et al. (2013)*. Validated shRNA sequences with the highest score for targeting mouse UBE2G1, UBE2G2, UBE2J1, and MARCH6 were selected from the shERWOOD database (www.sherwood.cshl.edu), and ~100 bp oligonucleotides with the corresponding sequence were synthesized. This sequence was amplified by PCR using degenerate primers with XhoI and EcoRI restriction sites. The PCR was digested with XhoI and EcoRI, gel purified, and ligated into the LT3GEPIR lentiviral vector for doxycycline-inducible RNAi expression with GFP reporter fluorescence. The LT3GEPIR vector was the kind gift of Johannes Zuber.

## Cell line generation

GFP-tagged cell lines were generated in C2C12 mouse myoblasts. Low-passage C2C12 cells were obtained from ATCC, and identity was validated by a functional assay: cells were grown to confluency, switched to low serum medium for several days, and evaluated for the formation of multinucleated myotubes. Parallel cultures of C2C12 cells were infected with virus encoding GFP fusion proteins. 293 T cells were transfected with delivery vectors and viral packaging vectors for retroviral or lentiviral production. Conditioned media were collected 48–72 hr after transfection and applied to low-passage C2C12 cells in the presence of polybrene. Integrated clones were selected using the

relevant antibiotic selections for each vector backbone. Fluorescent cell populations were isolated by FACS. The resulting stable GFP-expressing C2C12 cell lines were tested to verify the absence of mycoplasma contamination. miR-E RNAi cell lines were generated in U2OS cells. U2OS cells were obtained from ATCC and were periodically tested for mycoplasma contamination. EMDΔ95–99-GFP was introduced by retroviral infection and FACS sorted as described above. miR-E RNAi expression vectors were then introduced into these stable cell lines by lentiviral infection.

## RITE tag switching

RITE tag switching experiments were performed in quiescent C2C12 cells stably expressing RITE-tagged INM proteins. C2C12s were grown in Ibidi chamber slides and induced to enter quiescence as previously described (*Zhang et al., 2010*) by growing C2C12 myoblasts to ~ 75% confluence, washing twice in warm PBS, and switching to quiescent medium (DMEM without methionine, 2% FBS, and pen-strep). Cells were maintained in quiescent medium for 3 days with media changes every other day before initiation of RITE timecourses. To induce tag exchange, concentrated adenovirus expressing Cre recombinase was added to the culture medium. Tag switching was initiated at the indicated timepoints such that the entire slide containing all time points could be fixed, stained, imaged, and quantified in parallel. To quantify loss of 'old' RITE-tagged protein over time, intensity per unit area of the 'old' tag was quantified across all conditions. Background measurements were taken from cell-free regions of the imaging dish and subtracted from all signals as a background correction. All signals were then normalized to the day 0 timepoint (no tag switch).

## Antibody uptake assays

For antibody uptake assays, cells were pre-treated with drugs for the indicated times, then incubated in medium containing antibody, drug, and 125 µM leupeptin for 1 hr before fixation in paraformaldehyde and staining. Cells were stained with Alexa Fluor-conjugated secondary antibody to visualize internalized primary antibody:GFP conjugates. Cell surfaces were stained with Alexa Fluor-conjugated WGA; the WGA signal was used as a guide for outlining individual cells and quantifying internalized antibody fluorescence.

## Preparation of protein lysates and western blotting

Cells were washed in PBS, then lysed directly in plates in PBS lysis buffer (PBS supplemented with 1% Triton-X-100, 0.1% SDS, and protease inhibitors). Cells were further lysed by passage through a 25-gauge needle. Protein concentrations were quantified by BCA assay, and ~20 ug of total protein was loaded per lane of 4–12% gradient gels (Invitrogen). Blots were incubated with emerin antibody (1:1000) or alpha-tubulin antibody (1:5000) followed by IR Dye-conjugated secondary antibodies (1:5000) for multiplexed detection on the Odyssey imaging system.

## Microscopy and image analysis

Cells were grown in Ibidi culture chambers, treated as indicated, and fixed in 4% PFA for 5 min, then permeabilized in IF buffer (PBS, 0.1% Tx100, 0.02% SDS, 10 mg/ml BSA) before staining with Hoechst DNA dye. Wells were rinsed in PBS before imaging on a Zeiss LSM 710 scanning confocal microscope with a 63 × 1.4 NA objective. Images shown are single confocal slices. All image quantification was performed on maximum intensity projections of z-series with ImageJ. To quantify NE-localized protein levels, the DNA stain was used as a mask, and all GFP fluorescence within that mask was quantified.

For lysosomal staining, cells were prepared as described (*Castellano et al., 2017*) with the following modifications. Following fixation in 4% PFA for 5 min, cells were rinsed in PBS, then permeabilized in freshly prepared 0.1% digitonin in PBS for 10 min at 4C. Cells were rinsed again in PBS, then blocked in 2% goat serum in PBS for 30 min before staining with LAMP1 antibody (1:100 in 2% goat serum) for 1–2 hr at RT. Cells were rinsed again in PBS, then stained with Alexa Fluor-conjugated secondary antibody and Hoechst DNA stain for 1 hr at RT.

## Additional information

### Funding

| Funder | Grant reference number | Author |
|---|---|---|
| NIH Office of the Director | NS096786 | Martin Hetzer |
| National Institute of General Medical Sciences | R01GM126829 | Martin Hetzer |
| National Cancer Institute | P30 014195 | Martin Hetzer |
| Chapman Foundation | | Martin Hetzer |
| Helmsley Charitable Trust | | Martin Hetzer |

The funders had no role in study design, data collection and interpretation, or the decision to submit the work for publication.

### Author contributions

Abigail Buchwalter, Conceptualization, Data curation, Formal analysis, Validation, Investigation, Visualization, Methodology, Writing—original draft, Project administration, Writing—review and editing; Roberta Schulte, Data curation, Investigation, Methodology, Project administration; Hsiao Tsai, Investigation, Methodology; Juliana Capitanio, Software, Formal analysis, Methodology; Martin Hetzer, Resources, Supervision, Funding acquisition, Writing—review and editing

### Author ORCIDs

Abigail Buchwalter (iD) https://orcid.org/0000-0001-7181-6961

### Decision letter and Author response

Decision letter https://doi.org/10.7554/eLife.49796.030
Author response https://doi.org/10.7554/eLife.49796.031

## Additional files

### Supplementary files

• Source data 1. Filtered peptide data for half life calculations. Peptide turnover data for all peptides passing quality control filters. See R script and Materials and methods for details.
DOI: https://doi.org/10.7554/eLife.49796.021

• Source data 2. Filtered protein data for half life calculations. Filtered and averaged protein turnover data. See R script and Materials and methods for details.
DOI: https://doi.org/10.7554/eLife.49796.022

• Supplementary file 1. Results of half life fits passing quality filters.
DOI: https://doi.org/10.7554/eLife.49796.023

• Supplementary file 2. Complete list of half life fits.
DOI: https://doi.org/10.7554/eLife.49796.024

• Supplementary file 3. Half lives and protein topology data. Selected data related to *Figure 1G-H*.
DOI: https://doi.org/10.7554/eLife.49796.025

• Transparent reporting form  DOI: https://doi.org/10.7554/eLife.49796.026

### Data availability

Raw and analyzed mass spectrometric data and associated scripts and tables have been deposited in Dryad. Analyzed data are also included with the manuscript as supplementary tables.

The following dataset was generated:

| Author(s) | Year | Dataset title | Dataset URL | Database and Identifier |
|---|---|---|---|---|
| Buchwalter A, | 2019 | Data from: Selective clearance of | https://dx.doi.org/10. | Dryad Digital |

| Schulte R, Tsai H, Capitanio J, Hetzer MW | the inner nuclear membrane protein emerin by vesicular transport during ER stress | 5061/dryad.n0r525h | Repository, 10.5061/dryad.n0r525h |
| --- | --- | --- | --- |

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
