## [Decision Letter]

Thank you for submitting your article "Selective clearance of the inner nuclear membrane protein emerin by vesicular transport during ER stress" for consideration by *eLife*. Your article has been reviewed by three peer reviewers, including Elizabeth A Miller as the Reviewing Editor and Reviewer #1, and the evaluation has been overseen by David Ron as the Senior Editor. The following individuals involved in review of your submission have agreed to reveal their identity: Maurizio Molinari (Reviewer #2).

The reviewers have discussed the reviews with one another, and the Reviewing Editor has drafted this decision to help you prepare a revised submission.

This manuscript from the Hetzer lab uses SILAC pulse-chase mass spectrometry to measure the lifetime of nuclear envelope proteins, finding a range of turnover rates. For a short-lived protein, the authors go on to characterize the potential degradation pathways, finding no clear ubiquitination machinery that can be assigned to proteasome-mediated degradation, suggesting redundancy in this pathway. The authors then go on to characterize a second mode of degradation, mediated by vesicle trafficking machineries, and triggered by ER stress. This has hallmarks of the RESET pathway described by others and is exciting in presenting a new substrate for this pathway. The conclusions are largely supported by the data, and the description of a new RESET client is an exciting advance. It would have been nice to also define the proteasome-mediated pathway, but clearly this will require more dissection to overcome problems with redundancy.

The primary shared concern is over the use of chemical compounds (which have potential pleiotropic effects) over long time periods, with additional concern over the potential for gene expression effects to explain observed changes (e.g. Figure 3E). To this end, pulse-chase experiments are preferable for being quantitative and taking into account gene expression effects. Coupled with the glycosylation site addition (as suggested by reviewer 3), this should give a more detailed view of the fate of EMD under ER stress. We do not suggest that all experiments need to be repeated using this type of analysis, but a representative set of experiments covering the key conditions is required to strengthen the authors' model.

Reviewers also propose to more directly visualize the fate of EMD using the tag switching method to more acutely observe the transition of the "old" protein from the INM, to the ER, Golgi and then to the lysosome upon different treatments. A time course of imaging and co-localization would strengthen this argument. Indeed, some co-localization experiments are required even for the steady state observations, most notably for a lysosomal marker in the Baf A1 experiment. Similarly, a control IF experiment of unstressed cells treated with Baf A1 is required to demonstrate the requirement for ER stress in this remobilization.

*Reviewer #2:*

In this paper, the authors examine the intracellular fate of ectopically expressed emerin (EMD) and of a disease-causing variant thereof. EMD is used as model to investigate proteasomal and lysosomal pathways that regulate turnover of inner nuclear membrane (INM) proteins. They report that both wild type and mutant EMD are ERAD substrates. During acute ER stress, both wild type and mutant EMD are exported from the INM and the ER, via the Golgi and the plasma membrane, to the lysosomes for clearance.

There are few major issues to consider:

1) The authors monitor variations in intracellular level and localization of ectopically expressed EMD proteins in response to cell exposure to various compounds that inhibit ERAD, jeopardize ER to Golgi transport, induce ER stress, block protein synthesis. Note that all these compounds have pleiotropic effects and are to some extent toxic to cells at the concentrations and times (up to 24 hours) used in the experiments. Nevertheless, the authors solely ascribe these drug-induced variations in EMD's intracellular level and localization, to changes in EMD's turnover.

2) At this stage, it cannot be excluded that the various drugs (Figure 2E-G, Figure 3B-D, Figure 4…), or the induction of gene silencing (Figure 3E) modify the expression (rather than the clearance) of the EMDs. For example, it has been reported that MG132 enhances CMV promoter-regulated expression of ectopic genes (and EMDs gene expression is placed here under control of a CMV promoter). I suggest using Bortezomib, a more specific proteasome inhibitor. Also, in Figure 3E I notice that high levels of GFP are expressed only in cells were gene silencing has been activated. GFP expression could reduce expression of the second transgene (EMD), thereby offering an alternative explanation to the one proposed by the authors for the reduction of the EMD level upon E2 ligases knockdown.

All in all, the authors should systematically check synthesis of EMDs in their experiments and how EMDs synthesis changes under the experimental set-up. Moreover, they should measure EMDs stability directly, via quantitative methods such as pulse-chase analyses.

3) A major point of the paper (and the most interesting one) is that changes in cellular (ER) homeostasis trigger lysosomal clearance of proteins from the INM. I am not sure that this is (convincingly enough) supported by the results shown here. To demonstrate that EMD is cleared from the INM, the authors should monitor (by exploiting the epitope-exchange technology) the fate of "old" EMD and show that it re-localizes from the nuclear membrane, to the ER, Golgi and then to the lysosomes (i.e., old EMD should accumulate in the endo-lysosomes (=LAMP1-positive organelle) during ER stress, in the presence of Baf A1). Since EMDs are retained in the INM by association with Lamin A, is EMD:Lamin A complex regulated by ER stress?

4) I miss some experiment with endogenous proteins (e.g., EMD, Sun2, Lamin A). Is their turnover affected by ER stress? Are they delivered to endolysosomes upon ER stress induction?

*Reviewer #3:*

Buchwalter et al. investigate protein turnover in a mammalian cell system with a focus on proteins of the nuclear envelope and inner nuclear membrane (INM), an area of significant contemporary interest. In brief, the novelty of the manuscript lies in: (i) in the determination of half-lives of INM proteins in a tissue culture model of resting cells, hence minimizing the contribution of "canonical" ERAD via mixing of ER and INM through open mitosis; (ii) the application of RITE analysis in this context to also monitor protein localization, not only half-lives; (iii) chiefly the proposal of a novel route for degradation via trafficking through the Golgi-PM-lysosomal route and a definition a novel role for the LEM domain in this context.

In the opinion of this reviewer, the manuscript should be of considerable interest for the broad readership of *eLife*. While the identification of the responsible E3 ligase(s) should not be a key requirement for publication, a concern is that many of the key experiments rely solely on pharmacological inhibitors that often have pleiotropic/toxic effects, especially when used in combination. Some relatively straightforward experiments are suggested below that could help to strengthen the authors' proposal of a novel degradation route.

1) It is suggested to append a N-Glycosylation sequence (Asn-X-Ser/Thr) to the C terminus of emerin, a readout that is commonly used by laboratories studying tail-anchored protein biogenesis. This readout would be extremely useful for several reasons:

(i) inserted and preinserted "immature" variants and their degradative fate can be distinguished with ease on immunoblots;

(ii) trafficking from the INM or ER to and through the Golgi can be monitored with ease by monitoring the acquisition of Endo H resistance (vs. PNGase sensitivity), and;

(iii) most importantly, a glycosylation-competent variant is useful to reinforce the interpretation of trafficking to Golgi/lysosome: this EMD variant should accumulate as Endo H-resistant species upon lysosomal deacidification. Moreover, this observation would help to rule out ER-Phagy. In the opinion of this reviewer, it would not be necessary to repeat each and every experiment with this construct, but its application to a few key experiments would considerably strengthen the authors' proposal of a novel degradation route.

2) Can the authors rule out that only a selective sub-pool of "new" EMD variants enter the lysosomal pathway while another "old" subset is locally degraded? Perhaps the authors could consider demonstrating (via RITE) that "old", INM-resident EMD-variants are subject to lysosomal degradation? Alternatively, all formal possibilities could be stated/deconvoluted more clearly in the Discussion section.

---

## [Author Response]

This manuscript from the Hetzer lab uses SILAC pulse-chase mass spectrometry to measure the lifetime of nuclear envelope proteins, finding a range of turnover rates. For a short-lived protein, the authors go on to characterize the potential degradation pathways, finding no clear ubiquitination machinery that can be assigned to proteasome-mediated degradation, suggesting redundancy in this pathway. The authors then go on to characterize a second mode of degradation, mediated by vesicle trafficking machineries, and triggered by ER stress. This has hallmarks of the RESET pathway described by others and is exciting in presenting a new substrate for this pathway. The conclusions are largely supported by the data, and the description of a new RESET client is an exciting advance. It would have been nice to also define the proteasome-mediated pathway, but clearly this will require more dissection to overcome problems with redundancy.

We would like to thank the reviewers for their constructive comments. We have extensively reorganized the manuscript and included new data in response to the reviewers’ critiques. For clarity, we have reorganized the manuscript to focus mostly on the more rapidly degraded EMDΔ95-99 mutant. This makes it possible for us to show more data and controls for this mutant, while we show key comparisons with wild type EMD and the EMDΔLEM mutant. We feel that this more streamlined organization of the manuscript and the new data included with this revision strengthen the manuscript considerably.

The primary shared concern is over the use of chemical compounds (which have potential pleiotropic effects) over long time periods, with additional concern over the potential for gene expression effects to explain observed changes (e.g. Figure 3E).

We now include shorter timepoints of drug treatments (2-8 hours in many cases) which exhibit effects consistent with our observations at longer timepoints.

The reviewers expressed concern that over the time period of RNAi induction, expression of the free GFP encoded by the inducible miR-E RNAi vector might interfere with the synthesis of EMDΔ95-99-GFP. We have included supplemental data (Figure 3—figure supplement 2) showing that two methods of knockdown, siRNA transfection and miR-E RNAi induction, exhibit consistent effects. Neither siRNA transfection (Figure 3—figure supplement 2A-B) nor doxycycline-inducible miRNA expression (Figure 3C) stabilize EMDΔ95-99-GFP. In the siRNA transfection experiments, we observed some decrease of EMDΔ95-99-GFP levels in cells transfected with a scramble RNAi control, but no difference between this condition and specific knockdown conditions.

In the miR-E induction experiments, a GFP marker is co-expressed when the miRNA is induced. In some miR-E knockdown conditions (UBE2G1 or UBE2G2 miR-E, Figure 3C; MARCH6, Figure 3—figure supplement 2D), we observe loss of EMDΔ95-99-GFP. This could indicate that the protein is degraded by another pathway, or alternatively that protein synthesis is suppressed because of promoter competition between the EMDΔ95-99-GFP and the free GFP. We think the latter interpretation is unlikely because an additional knockdown (UBE2J1, Figure 3—figure supplement 2) also had no effect on emerin-GFP levels, even though free GFP is also expressed in this condition. These siRNA and miR-E experiments are overall unified in their outcome: knockdown of individual ERAD-implicated E2 or E3 ubiquitin ligases does not cause emerin-GFP to accumulate within cells.

We agree that it is likely that multiple E2 and/or E3 ligases are redundant in ERAD-mediated degradation of EMD. However, these results were an initial hint that an alternative pathway for EMD degradation exists. We acknowledge this possible redundancy in subsection “Proteasome-dependent and proteasome-independent modes of emerin clearance”:

*“*We depleted MARCH6, Rnf26, and CGRRF1 with siRNA, but observed no effect on EMDΔ95-99 protein levels, suggesting that these ligases do not catalyze EMD turnover, or alternatively that multiple E3 ligases are redundant in this process.”

To this end, pulse-chase experiments are preferable for being quantitative and taking into account gene expression effects. Coupled with the glycosylation site addition (as suggested by reviewer 3), this should give a more detailed view of the fate of EMD under ER stress. We do not suggest that all experiments need to be repeated using this type of analysis, but a representative set of experiments covering the key conditions is required to strengthen the authors' model.

We now include RITE time courses for both wild type EMD and EMDΔ95-99, which show that EMDΔ95-99 disappears more rapidly than wild type EMD, but that both variants are stabilized by proteasome inhibition at the NE (Figure 2E-H).

We also include a cycloheximide time-course showing the relative stabilities of wild type EMD-GFP, EMDΔ95-99-GFP (Figure 3—figure supplement 1C-D), and EMDΔLEM-GFP (Figure 8—figure supplement 1). Importantly, deletion of the LEM domain makes EMD more stable, consistent with our model that the LEM domain mediates targeting of emerin variants to the lysosome.

We now also include a cycloheximide + thapsigargin time-course of EMDΔ95-99-GFP which indicates that under ER stress, EMDΔ95-99-GFP disappears more quickly and is detectable in a higher molecular weight modified form (Figure 3D).

Reviewers also propose to more directly visualize the fate of EMD using the tag switching method to more acutely observe the transition of the "old" protein from the INM, to the ER, Golgi and then to the lysosome upon different treatments.

Unfortunately, the RITE tag switching method lacks the temporal control needed to dissect induced turnover of “old” protein in response to ER stress on timescales of minutes to hours. This is because the tag switching event involves first editing of the plasmid by Cre recombinase within cells, then production of new RNA and finally protein. In the meantime, any remaining RNA that encodes the “old” protein will remain in the cell until it degrades. These factors introduce an inevitable lag time in turnover and make it difficult to pinpoint a clearly defined “old” population of protein for tracking on short timescales.

Using stable cell lines that express GFP-tagged EMD variants at roughly endogenous levels, we observe that NE-localized EMD rapidly decreases, while the protein enriches in the Golgi (Figure 4), is detectable at the plasma membrane (Figure 6) and is found in the lysosome (Figure 5) over time. We infer from this that a significant proportion of EMD leaves the INM and moves through the secretory pathway upon ER stress, but we cannot rule out at this point that some of the “old” protein, or maturely folded protein, is degraded in situ at the INM.

A time course of imaging and co-localization would strengthen this argument. Indeed, some co-localization experiments are required even for the steady state observations, most notably for a lysosomal marker in the Baf A1 experiment.

Figure 4 shows time- and stress-dependent colocalization of EMDΔ95-99-GFP with the Golgi resident protein giantin. We now include costaining with the lysosomal protein LAMP1, which clearly shows that emerin accumulates in lysosomes when lysosome acidification is blocked by bafilomycin A1 treatment (Figure 5D-F). LAMP1, which marks the limiting membrane of the lysosome, can clearly be seen encircling the EMDΔ95-99-GFP signal within the interior of the lysosome.

Similarly, a control IF experiment of unstressed cells treated with Baf A1 is required to demonstrate the requirement for ER stress in this remobilization.

We now include this data in Figure 5E; EMDΔ95-99-GFP does not enrich in LAMP1-marked lysosomes without ER stress induction.

Reviewer #2[…]2) At this stage, it cannot be excluded that the various drugs (Figure 2E-G, Figure 3B-D, Figure 4…), or the induction of gene silencing (Figure 3E) modify the expression (rather than the clearance) of the EMDs. For example, it has been reported that MG132 enhances CMV promoter-regulated expression of ectopic genes (and EMDs gene expression is placed here under control of a CMV promoter). I suggest using Bortezomib, a more specific proteasome inhibitor. Also, in Figure 3E I notice that high levels of GFP are expressed only in cells were gene silencing has been activated. GFP expression could reduce expression of the second transgene (EMD), thereby offering an alternative explanation to the one proposed by the authors for the reduction of the EMD level upon E2 ligases knockdown.All in all, the authors should systematically check synthesis of EMDs in their experiments and how EMDs synthesis changes under the experimental set-up. Moreover, they should measure EMDs stability directly, via quantitative methods such as pulse-chase analyses.

We now include two types of pulse-chase experiments which indicate that EMD is stabilized by proteasome inhibition. In Figure 2, we use the RITE system to directly track the localization and levels of “old” EMD variants in the absence or presence of MG132 and observe that MG132 treatment stabilizes “old” EMD. Secondly, we include representative images of cells stably expressing EMDΔ95-99-GFP and treated with CHX, MG132, or CHX + MG132 (Figure 2—figure supplement 1A). This indicates that MG132 inhibits degradation of mature protein (CHX + MG132 condition). We also observe a widespread increase in EMDΔ95-99-GFP signal throughout cell compartments in the + MG132 alone condition. In this condition, it is possible that MG132 is enhancing expression of EMDΔ95-99-GFP. We think it is more likely that significant amounts of EMDΔ95-99-GFP are cotranslationally degraded, as is known to occur for misfolded tail-anchored proteins (Hessa and Hegde, Nature, 2011). We also observe that MG132 treatment causes the accumulation of higher molecular weight, likely poly-ubiquitinated variants of EMDΔ95-99-GFP (Figure 3C), which we interpret as stalled degradation rather than increased synthesis of the protein.

3) A major point of the paper (and the most interesting one) is that changes in cellular (ER) homeostasis trigger lysosomal clearance of proteins from the INM. I am not sure that this is (convincingly enough) supported by the results shown here. To demonstrate that EMD is cleared from the INM, the authors should monitor (by exploiting the epitope-exchange technology) the fate of "old" EMD and show that it re-localizes from the nuclear membrane, to the ER, Golgi and then to the lysosomes (i.e., old EMD should accumulate in the endo-lysosomes (=LAMP1-positive organelle) during ER stress, in the presence of Baf A1). Since EMDs are retained in the INM by association with Lamin A, is EMD:Lamin A complex regulated by ER stress?

This manuscript shows that a protein that localizes to the INM can also be targeted to the lysosome, which is a surprising and novel finding. We observe that INM-localized EMDΔ95-99-GFP is undetectable at the NE after ~8 hours of ER stress (Figure 3E-F). This clearly indicates that EMDΔ95-99 is quantitatively degraded during ER stress. However, we also see evidence that EMD can be degraded by a proteasome-dependent pathway under some conditions, and we think it is likely that this represents ERAD. We cannot definitively rule out the possibility that some proportion of EMD is degraded by each of these pathways during ER stress. Defining the spectrum of EMD interactions in normal vs. stressed cells will likely help to illuminate this, and future work will address this important question. We do know that disruption of the EMD:lamin A complex is not sufficient to cause EMD to be degraded by the lysosome, as we see in *lmna* -/- MEFs that EMD-GFP is still stably expressed but loses its affinity for the INM and instead localizes to the peripheral ER (Figure 8—figure supplement 2). Rather, our data suggest that the LEM domain:BAF interface is likely to be the binding interaction that is regulated by ER stress.

4) I miss some experiment with endogenous proteins (e.g., EMD, Sun2, Lamin A). Is their turnover affected by ER stress? Are they delivered to endolysosomes upon ER stress induction?

Our explorations so far suggest that endogenous emerin is not quantitatively degraded during ER stress, although its levels do modestly increase at the NE after brefeldin A treatment. This could be consistent with our LEM domain competition model (see Figure 9 and Discussion); if emerin is expressed more highly than its nucleoplasmic BAF binding partner, more of emerin’s LEM domains may be unbound to BAF and may be able to associate with other factors, including factors that might enable ER export. However, we think it is possible that endogenous wild type or disease-mutant emerin may flux through this pathway at some lower level under certain conditions, and future experiments will explore this possibility.

Reviewer #31) It is suggested to append a N-Glycosylation sequence (Asn-X-Ser/Thr) to the C terminus of emerin, a readout that is commonly used by laboratories studying tail-anchored protein biogenesis. This readout would be extremely useful for several reasons:(i) inserted and preinserted "immature" variants and their degradative fate can be distinguished with ease on immunoblots;(ii) trafficking from the INM or ER to and through the Golgi can be monitored with ease by monitoring the acquisition of Endo H resistance (vs. PNGase sensitivity);(iii) most importantly, a glycosylation-competent variant is useful to reinforce the interpretation of trafficking to Golgi/lysosome: this EMD variant should accumulate as Endo H-resistant species upon lysosomal deacidification. Moreover, this observation would help to rule our ER-Phagy. In the opinion of this reviewer, it would not be necessary to repeat each and every experiment with this construct, but its application to a few key experiments would considerably strengthen the authors' proposal of a novel degradation route.

We would like to thank this reviewer for this suggestion, which has provided important supporting data for this revised manuscript. We have generated glycosylation reporter cell lines for EMDΔ95-99-GFP* (Figure 3H, Figure 5G), EMD-WT-GFP* (Figure 7E-F), and EMDΔLEM-GFP* (Figure 8) and performed (i) THG and CHX co-treatments and (ii) THG and Baf A1 cotreatments with each of these lines. We tracked the response of a pool of mature protein during ER stress by co-treating cells with cycloheximide to block new protein synthesis, and thapsigargin to induce ER stress. These experiments clearly indicate that within 2 hours of ER stress induction EMDΔ95-99-GFP* (Figure 3G-H) and EMD-WT-GFP* (Figure 7E) shift from predominantly Endo H-sensitive species to predominantly Endo H-resistant species. This indicates that these proteins progressively leave the ER and enter post-ER compartments.

In a short 4 hour THG + Baf A1 treatment time-course, the Endo H-resistant pool of these proteins also increases over time (EMDΔ95-99-GFP*, Figure 5G; EMD-WT-GFP*, Figure 7F). The conversion from Endo H-sensitive to Endo H-resistant is not as quantitative here, likely because we did not include inhibition of protein synthesis along with these two other treatments. Nevertheless, Endo H-resistant species become more abundant when lysosome acidification is inhibited along with ER stress induction.

Both EMD-WT and EMDΔ95-99 are partially Endo H-resistant under homeostatic conditions. This likely indicates that these proteins are exiting the NE/ER with some frequency and potentially being degraded by the same lysosome-dependent pathway. Importantly, however, EMDΔLEM-GFP* shows a distinct phenotype. EMDΔLEM-GFP* exists in only one modified species that is Endo H-sensitive and does not become Endo H-resistant during ER stress (Figure 8I-J). This outcome indicates, in line with our other data, that the LEM domain is required for EMD variants to exit the NE/ER network and target to lysosomes for degradation.

2) Can the authors rule out that only a selective sub-pool of "new" EMD variants enter the lysosomal pathway while another "old" subset is locally degraded? Perhaps the authors could consider demonstrating (via RITE) that "old", INM-resident EMD-variants are subject to lysosomal degradation? Alternatively, all formal possibilities could be stated/deconvoluted more clearly in the Discussion section.

We infer from our data that a significant proportion of EMD leaves the INM and moves through the secretory pathway upon ER stress, but we cannot rule out at this point that some of the “old” protein, or maturely folded protein, is degraded in situ at the INM. We now address this in the Discussion section:

“While our data indicate that a significant proportion of EMD leaves the NE/ER during ER stress, we cannot rule out the possibility that ERAD-mediated degradation of some proportion of EMD takes place within the NE/ER network in parallel to the lysosome-mediated pathway that we have identified.”